# The ROCEEH Out of Africa Database (ROAD): A large-scale research database serves as an indispensable tool for human evolutionary studies

Andrew W. Kandel[1]*, Christian Sommer[1], Zara Kanaeva[1], Michael Bolus[1,2‡], Angela A. Bruch[3‡], Claudia Groth[3], Miriam N. Haidle[2,3], Christine Hertler[3‡], Julia Heß[3], Maria Malina[1‡], Michael Märker[4,5‡], Volker Hochschild[6], Volker Mosbrugger[3], Friedemann Schrenk[3], Nicholas J. Conard[2]

1 The Role of Culture in Early Expansions of Humans, Heidelberg Academy of Sciences and Humanities, Tübingen, Germany, 2 Department of Geosciences, Working Group Early Prehistory and Quaternary Ecology, University of Tübingen, Tübingen, Germany, 3 The Role of Culture in Early Expansions of Humans, Senckenberg Forschungsinstitut, Heidelberg Academy of Sciences and Humanities, Frankfurt/Main, Germany, 4 Department of Earth and Environmental Sciences, University of Pavia, Pavia, Italy, 5 Working Group on "Soil Erosion and Feedbacks", Leibniz Centre for Agricultural Landscape Research (ZALF), Müncheberg, Germany, 6 Institute of Geography, Department of Geosciences, University of Tübingen, Tübingen, Germany

☯ These authors contributed equally to this work.
‡ MB, AAB, CH, MM and MM also contributed equally to this work.
* a.kandel@uni-tuebingen.de

## Abstract

Large scale databases are critical for helping scientists decipher long-term patterns in human evolution. This paper describes the conception and development of such a research database and illustrates how big data can be harnessed to formulate new ideas about the past. The Role of Culture in Early Expansions of Humans (ROCEEH) is a transdisciplinary research center whose aim is to study the origins of culture and the multifaceted aspects of human expansions across Africa and Eurasia over the last three million years. To support its research, the ROCEEH team developed an online tool named the ROCEEH Out of Africa Database (ROAD) and implemented its web-based applications. ROAD integrates geographical data as well as archaeological, paleoanthropological, paleontological and paleobotanical content within a robust chronological framework. In fact, a unique feature of ROAD is its ability to dynamically link scientific data both spatially and temporally, thereby allowing its reuse in ways that were not originally conceived. The data stem from published sources spanning the last 150 years, including those generated by the research team. Descriptions of these data rely on the development of a standardized vocabulary and profit from online explanations of each table and attribute. By synthesizing legacy data, ROAD facilitates the reuse of heritage data in novel ways. Database queries yield structured information in a variety of interoperable formats. By visualizing data on maps, users can explore this vast dataset and develop their own theories. By downloading data, users can conduct further quantitative analyses, for example with Geographic Information Systems, modeling programs and artificial intelligence. In this paper, we demonstrate the innovative nature of

**Data Availability Statement:** All relevant data for this study are publicly available from the Zenodo

repository (https://doi.org/10.5281/zenodo.7669784).

**Funding:** NJC, VH, VM, FS are the Principal Investigators of the ROCEEH research center. The ROCEEH research center of the Heidelberg Academy of Sciences and Humanities (https://www.hadw-bw.de/) was promoted by the Joint Science Conference of the Federal Government and the state governments of the Federal Republic of Germany in the Academies' Programme of the Union of the German Academies (https://www.akademienunion.de/forschung/akademienprogramm/). Since 2008, ROCEEH and the research and development of ROAD have been generously supported by the Federal Government of Germany (Federal Ministry of Education, Science and Research) as well as the states of Baden-Württemberg (Ministry of Science, Research and the Arts) and Hesse (Ministry of Science and the Arts). The funders had no role in study design, data collection and analysis, decision to publish, or preparation of the manuscript.

**Competing interests:** The authors have declared that no competing interests exist.

ROAD and show how it helps scientists studying human evolution to access datasets from different fields, thereby connecting the social and natural sciences. Because it permits the reuse of "old" data in new ways, ROAD is now an indispensable tool for researchers of human evolution and paleogeography.

## 1. Introduction

This paper describes the conception and development of a large-scale research database covering the topic of human evolution. The first parts provide background about the project and details the technical aspects considered by our team to implement the database. The next part addresses the contents of the database to provide an overview of its holdings. Finally, we describe some of the functions and applications that make the database indispensable for scientists studying human evolution and examine its potential beyond the project's lifetime.

The Role of Culture in Early Expansions of Humans (ROCEEH) is a twenty-year (2008–2027) research center of the Heidelberg Academy of Sciences and Humanities operating within the framework of the Academies' Programme of the Union of the German Academies. With a team well versed in the study of archaeology, paleoanthropology, paleobiology, paleoenvironment and paleogeography, ROCEEH explores the history of humanity and its early expansions between three million and 20,000 years ago in Africa and Eurasia.

ROCEEH recognizes human evolution as a story of expansions. During the last two million years, the genus *Homo* spread from Africa into Asia and Europe in several waves of dispersal. New species developed and old groups became extinct during these *range expansions*. As early as three million years ago, hominins had established new ways of dealing with their specific environment through culture. New forms of tools opened up access to new resources and activated changes in body, mind and behavior, resulting in the *expansion of performances*. Changes in the ecospace of hominin species affected the viability and development of potential resources through natural processes, as these were also increasingly shaped by cultural activities, as reflected in *expansions of resource space*.

ROCEEH's mission is to develop a systemic understanding of "becoming human", one which integrates these three types of expansion and how they interacted with each other. The aim of the project is to discover, contextualize and preserve information about the deep past of humankind's cultural heritage. ROCEEH makes this information accessible by compiling data about archaeological sites and their associated assemblages in its multidisciplinary, web-based, geo-relational database known as the ROCEEH Out-of-Africa Database (ROAD). With its Geographical Information Systems functionality, ROAD unifies spatial data about sites with additional information about the chronostratigraphical structure of layers and the archaeological finds those layers contain. In addition, ROAD assimilates further information about human fossils, fauna, flora and climate—information which can be used to model early human habitats.

Archaeology and its allied fields have increasingly embraced the importance of creating, sharing and reusing data, as these tend to drive scientific progress in the digital age [1]. Digitalization, coupled with an attitude towards open science and FAIR (Findable, Accessible, Interoperable, Reusable) sharing, preserves information from inevitably destructive research practices [2, 3], ensures reach, comprehensiveness and longevity [4], and facilitates sustainable access to information [5]. This best practice is increasingly implemented by researchers themselves. It is also supported by general trends in research [6, 7], especially when funding agencies require that a Data Management Plan accompany awarded grants [8].

In order to reuse scientific information effectively [9, 10], data need to be structured and make use of a nomenclature accepted by many stakeholders. Three examples from relatively new and related fields provide potential models illustrating how this approach can succeed. First, the field of human paleogenetics has adopted an outstandingly high degree of data sharing [11] which leads to the development of comprehensive, open access databases [12] and the ability for researchers to analyze unified datasets [13]. Second, in the realm of chronometric dating, the combination of assembled datasets [14] and refined analytical methods [15] allows for new insights to be derived from existing data. Third, an example from paleoenvironmental studies is the Neotoma database, whose centralized infrastructure serves as a go-to point for researchers who share data, and which simultaneously establishes a de-facto standard for structuring data [16].

Many aspects of standardization have been widely discussed in the field of archaeology, including digitalization of primary data [17–19], definition of metadata, ontologies and vocabularies [20, 21], and modes of reuse [22]. These factors have led to the development of institutions in the public domain (e.g. Archaeology Data Service, York) and private sectors (e.g. The Digital Archaeological Record, Tempe) who strive to preserve archaeological data. Furthermore, data aggregators like ARIADNEplus (Prato) seek to overcome the fragmentation of diverse national data providers by making the metadata accessible through a unified interface [23].

In this landscape of research infrastructures, we note two deficits that the ROCEEH research center aims to fill with its database. The first relates to the availability of information published before the digital revolution and information not published in a machine-actionable way [24]. For this purpose, we developed and executed a workflow to extract information from diverse sources with a high degree of quality control. The second relates to a lack of lateral connectedness between scientific disciplines. The aforementioned examples demonstrate best practice in integrating information from their respective fields. However, to get a systemic understanding of the human past, we require synthetic datasets, ones that comprise information from diverse fields and facilitate collaborative research [25]. ROCEEH developed a database structure that reflects the research center's interdisciplinary approach. By incorporating cultural, anthropological, paleoenvironmental and geographical data into a single framework, we had to harmonize the inherent conceptual models from a diverse set of disciplines related to human evolution, bridge taxonomic differences, and formalize the conventions in a common database design.

## 2. What is ROAD? Structure, implementation and data

ROAD is a structured collection of information organized so that data can be easily accessed, managed and updated with a database application. As we developed ROAD, it became clear that it would not consist of just a handful of tables. Using the original grant proposal as a source of inspiration, the team created a robust storehouse of data related to human evolution which could serve as a valuable research tool for the scientific community. The project database reflects published scientific literature as correctly and completely as possible. Thus, the use of the database was conceived in broad terms because we expected that investigations related to the research center's main focus—studying the role of culture in early expansions of humans—would generate further questions that necessitated expanding the database.

Our search for a suitable database system led to the implementation of a relational database whose main advantage is that users are free to join and manipulate data in ways that its designers did not foresee. This relational database can be easily expanded through the addition of new tables and attributes. Its creation relied on two main requirements: 1) the development of

a logical model to define the entities and attributes, as well as their relationships; and 2) the implementation of a logical model, including objects such as tables (which correspond to defined entities) and columns (which correspond to the attributes of the defined entities) [26].

We developed the logical part of the ROAD model using the visual tool called MySQL Workbench to create an Entity Relationship Diagram (S1 Fig). The model depicts how entities relate to each other, and which attributes describe which entity. Since archaeological and paleobiological datasets use varying terminologies, the ROCEEH team had to select appropriate and unique names from multiple options for each of the entities and attributes in the ROAD model. For example, we chose the attribute name 'locality' over 'site' or 'find-spot' and decided on 'assemblage' instead of 'find' or 'sample'. Regardless of what type of assemblage is represented, ROAD's Entity Relationship Diagram models the following simplified story: "Assemblages are found in localities; assemblages lie in geological and archaeological layers. Geological and archaeological layers are dated by one or more laboratories. Assemblages can also be dated by one or more laboratories. Assemblages have various descriptions." Furthermore, the logical model considers the publications used to generate the data.

ROAD enables user interaction through its application called ROADWeb. User interaction includes: entering, updating, and querying data using Structured Query Language (SQL). Additionally, users can migrate table data or query results from the database format to another interoperable one.

## 2.1 General overview of ROAD structure

ROAD consists of more than 50 tables organized into seven general themes: 1) localities; 2) geological layers and profiles; 3) archaeological layers and profiles; 4) dating results; 5) assemblage descriptions and derived data; 6) bibliographic information; and 7) linking tables. Fig 1 illustrates the conceptual structure of ROAD representing both the logical model and its implementation. In the logical model, the rectangles represent the general themes contained in the database, while the central oval and the lines connecting it to the general themes depict relationships. Because the rectangles depict one or many entities, most of the connecting lines represent more than one relationship. All lines joining the central oval represent many-to-

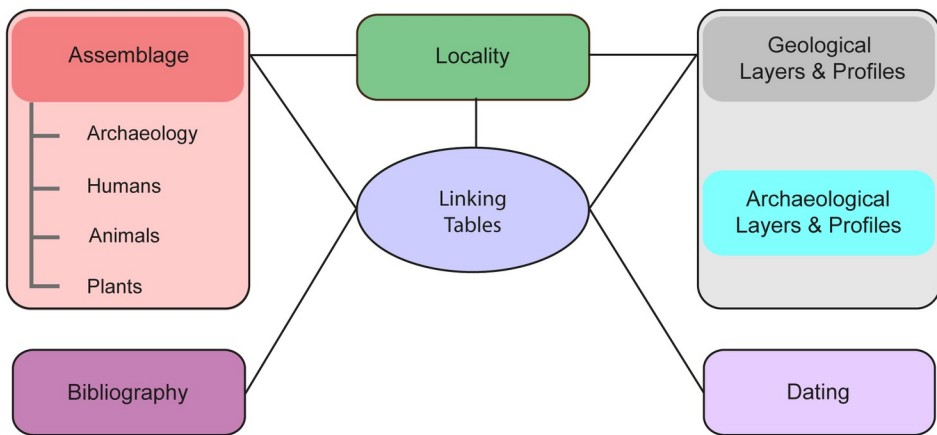

**Fig 1. The conceptual structure of ROAD.** The rectangles represent the generalized themes of the database. The central oval and lines connecting it to the themes depict relationships. Because the rectangles depict one or many entities, most of the lines represent more than one relationship. All lines joining the central oval represent many-to-many relationships. The other lines represent one-to-many relationships.

many relationships, while the other lines represent one-to-many relationships. The ROAD Table Descriptions (S1 File) provide detailed definitions of all tables and their attributes, specifications for data entry and examples of possible entries.

The implementation of the logical model involved seven groups of tables, which we outline here. The **first** group centers on the table "locality" which contains general data about the sites. The **second** group includes tables with information about the geological layers, which when viewed together represent geological profiles. Similarly, the **third** group includes tables with information about the archaeological layers, which when viewed together represent archaeological profiles. The **fourth** group consists of tables related to the ages of the layers and finds and contains radiometric dating results. The **fifth** group revolves around the table "assemblage" and incorporates tables with detailed descriptions of the assemblages. We will describe these five groups more fully in the following sections.

The **sixth** group of tables records bibliographic information. References are stored in a structured way and linked so that users can cite the sources of information contained in the previous five groups. The bibliographic information is contained in three separate tables: "publication_source", usually reflecting a book or a journal; "edition", containing the year, volume and issue number; and "publication", detailing title, authors and page numbers.

The **seventh** and last group of tables includes three generalized types of linking tables which implement many-to-many relationships. First, tables with the prefix "publication_-desc") link the table "publication" to one of the following tables: locality, assemblage, geological_layer, archaeological_layer, geological_stratigraphy, archaeological_stratigraphy, assemblage, humanremains, paleofauna, climate and vegetation. These tables join the publication to the specified records. Second, some tables join elements to establish specified relationships: 1) geological layers to geological stratigraphy; 2) archaeological layers to archaeological (cultural) stratigraphy; 3) archaeological layers to correlating archaeological or geological layers; and 4) assemblages to geological or archaeological layers. Third, some tables contain fixed lists of entries such as the taxonomic classification of hominins, animals and plants, lists of continents, countries and regions, and the definitions of archaeological cultures. These lists serve as look-up tables with drop-down menus to facilitate the standardized entry of data. A summary of tables associated with each of the four scientific disciplines in ROAD is presented in S1 Table.

## 2.2 Localities, layers and dating information

In ROAD, data entry for a new locality begins with the table "locality", as this serves as a primary key. Fig 2 shows the list of attributes contained in the table "locality". This table provides information about the geographic location of a site and the precision of the coordinates (e.g. unchecked, checked, landmark, GPS). It also describes the type of site (e.g. cave, rockshelter, open-air) and presents a text of up to 600 characters summarizing important aspects of the locality.

After locality, data entry continues with the geological and archaeological layers. The table "geological_layer" contains data including sediment type and thickness. The geological layers are used to generate a geological profile for every locality in the database. Similarly, the table "archaeological_layer" contains data about each layer's cultural affinity and is used to generate an archaeological profile for each locality containing archaeological finds. Fig 3 shows the attribute list of the table "geological_layer" as well as a dynamically generated geological profile for the locality "Rhafas Cave".

In ROAD, information about "age" is stored in three separate tables: "geological_layer_-age", "archaeological_layer_age" and "assemblage_age". The three age tables include

| locality |
| --- |
| idlocality<br>country<br>region<br>type<br>x<br>y<br>z<br>coordinate_source<br>no_data_entry<br>figure<br>comments<br>owner<br>created |

**Fig 2. The table "locality" and its attributes.** In addition to the attributes shown, all tables in ROAD include fields for comments, the name of the owner, the date the record was created, and a history of modification. The ROAD Table Descriptions (S1 File) provide detailed definitions of all tables and their attributes, specifications for data entry and examples of possible entries.

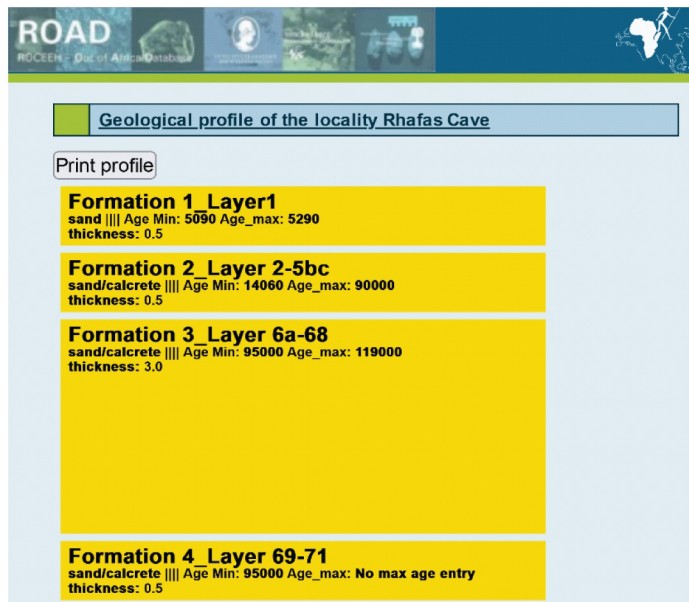

| geological_layer |
| --- |
| locality_idlocality<br>name<br>geoprofile_name<br>sediment<br>description<br>thickness<br>environment<br>micromorphology<br>overlying_geolayer_idlocality<br>overlying_geolayer_name<br>comments<br>owner<br>created |

**Fig 3. The table "geological layer" and its attributes (left) and a dynamically generated geological profile for the locality Rhafas Cave (right).** Rhafas Cave consists of a single profile which contains four geological layers. Each of the visualized layers depicts information about its name, sedimentology, thickness and age. By clicking on a geological layer, a user can find out more information about the corresponding archaeological layers and the assemblages they contain.

individual radiometric dating results from layers and assemblages. The layers and assemblages can have many dates and these can be based on different dating methods. It is important to note that these tables contain lists of raw data without interpretation. Therefore, the age information cannot be used alone to conduct age queries.

To run reliable age queries, we use the data in the three age tables to develop a static age model. The resulting age models for each layer are stored in the table "geological_stratigraphy". The age model presents our interpretation, which consists of a minimum and maximum age for each geological layer. This interpretation is displayed in the geological profile and applied to all automatically generated age queries. We base our interpretation not only on radiometric dating results, but also on relative dating, while taking geological, archaeological and stratigraphical considerations into account. For this reason, we specify the reasons for our determinations in an attribute named 'correlation' (e.g. radiocarbon dating, culture, stratigraphic relationship, uranium-series dating or magnetostratigraphy).

Furthermore, the table "archaeological_stratigraphy" ascribes age ranges to each cultural entity, for example, the Aurignacian in Europe spans from ca. 43,000 to 32,000 years ago, while the Levantine Aurignacian in southwestern Asia ranges from ca. 40,000 to 30,000 years ago. By establishing such definitions, we further constrain the ages of the geological and archaeological layers.

## 2.3 Assemblages in ROAD

Since ROCEEH is transdisciplinary by nature, assemblages stored in ROAD include not just archaeological finds, but also human, faunal and plant remains. Once a locality and its geological and archaeological layers are entered in ROAD, the table "assemblage" can be completed. Fig 4 lists the attributes of the table "assemblage", and Fig 5 shows the general classification of assemblages stored in ROAD. Two of the attributes serve to score the quality of an assemblage based on a scale from zero to three. The attribute 'is_systematic' denotes the degree to which an assemblage was collected systematically and spans from: (0) unknown, (1) surface collection without context, (2) survey with general coordinates, and (3) excavated with piece plotting. The attribute 'collection_bias' examines the representativeness of an assemblage and ranges from (0) unknown, (1) exchange of specimens, (2) well documented but incomplete, and (3) well documented and complete.

**2.3.1 Detailed assemblage description: Archaeological finds.** As shown in Fig 5, the category "archaeological finds" is divided into subcategories: stone artifacts, organic tools, symbolic artifacts, features and miscellaneous finds. The stone artifacts, or lithics, are further divided into four groups: raw material, typology, technology and function (Fig 6). In general, all archaeological finds are entered as groups of finds and not as single artifacts.

The first lithic table "raw_material" (Fig 7) provides specific information about different types of rocks present (e.g. chert, quartzite, obsidian, basalt). Since hominins gathered these rocks at specific places in the landscape and brought them to a locality to manufacture stone tools, we first classify the raw material based on its distance of transport from source to locality. The five classes of 'transport distance' are: local (0–5 km), regional (6–20 km), supra-regional (21–100 km), distant (>100 km) and unknown. While entering data, a user can select different rock types from the attribute 'raw_material_list' or add new raw materials. The percentage of raw materials is recorded for each transport distance with respect to the total number of artifacts present in the entire assemblage.

The second lithic table "typology" provides information about the types of stone artifacts present at a locality. Typology distinguishes mainly chipped and unchipped stone tools, whereas cores and debitage are included as non-tool forms. We define chipped tools as

**assemblage**

locality_idlocality
idassemblage
name
category
lithic_piece_count
coordinates_present
x
y
z
is_systematic
collection_bias
date
comments
owner
created

**Fig 4. The table "assemblage" and its attributes.** The first two attributes specify the locality and a unique identification number for each assemblage at that locality. The attribute 'name' contains a unique name that identifies the assemblage, e.g. Layer VI lithics or Horizon B human remains. The attribute 'category' lists which tables in ROAD contain further information about the assemblage. If an assemblage consists of stone artifacts, the 'lithic piece count' reflects the total number of artifacts contained in that assemblage. The degree to which an assemblage was collected systematically is scored (0–3) according to the sampling procedure, while its representativeness (0–3) is scored to consider any bias in collection.

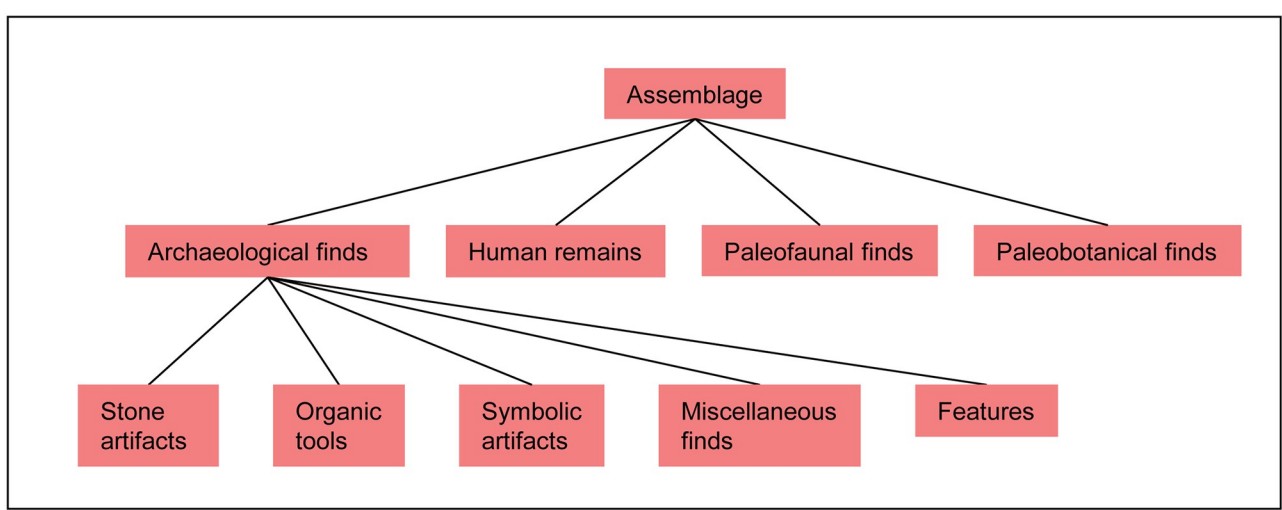

**Fig 5. Assemblages stored in ROAD are categorized into four basic classes.** The classes include archaeological finds as well as human, fauna and plant remains. The archaeological finds are further divided into subclasses for stone artifacts, organics tools, symbolic artifacts, miscellaneous finds and features.

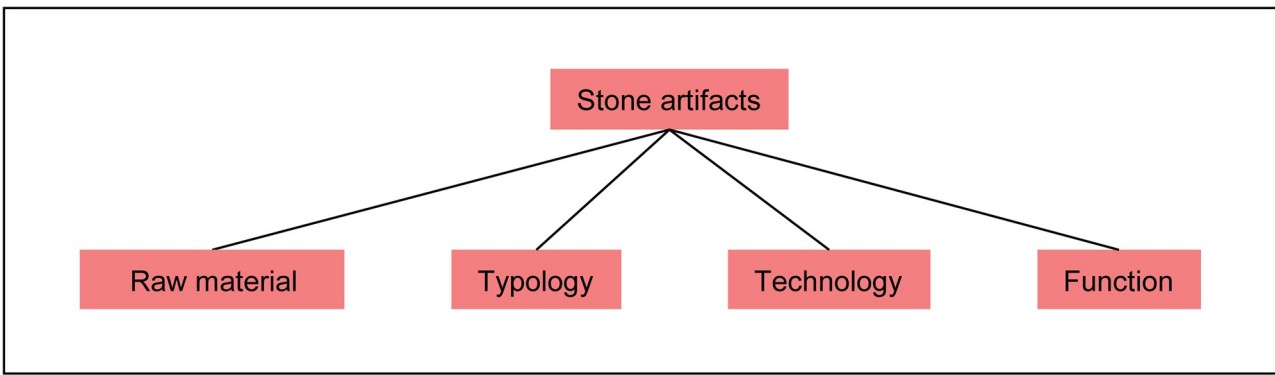

**Fig 6. Classification of stone artifacts.** Lithic assemblages are divided into raw material, typology, technology and function.

intentionally retouched products (e.g. scraper, backed blade or drill), while unchipped tools—generally known as groundstone tools—achieve their form mainly through use (e.g. hammerstone, grinder or anvil). Thus, the four basic categories are: chipped tool, non-chipped tool, non-tool and unknown. While entering data, a user can select different artifact types from the attribute 'tool_list' or add new types of artifacts. For each basic category, the percentage is recorded with respect to the total number of artifacts present in the assemblage. Ornaments or art objects made of stone are not considered here, but instead appear in the table "symbolic_artifacts".

The third lithic table "technology" differentiates how stone artifacts were manufactured. Technology encompasses mainly cores, blanks and debitage, although chipped and non-chipped tools are also of interest. The four categories of technology are: chipping technology, non-chipping technology, manuport and unknown. While entering data, a user can select different artifact types from the attribute 'product_list' and different technologies from the attribute 'technology_type'; new products and technologies can also be added. The percentage of artifacts present for each category is recorded with respect to the total number of artifacts present in the entire assemblage.

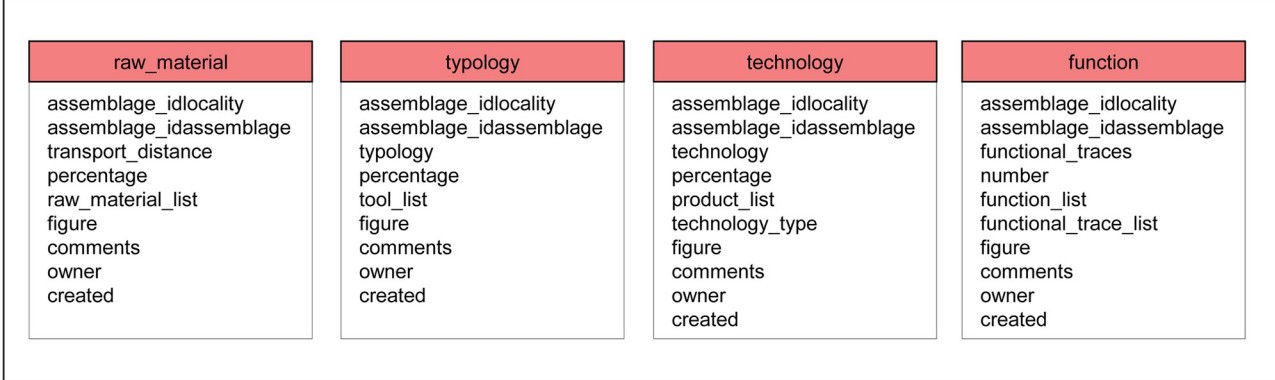

**Fig 7. Tables for analyzing stone artifacts listing the attributes of each table.** Raw materials include the distance to source, while typology describes the types of tools, cores and debitage. Technological characteristics of the stone artifacts (e.g. Levallois or Kombewa) and the results of functional analyses (e.g. use wear or residue studies) can also be entered.

The fourth lithic table "function" stores information about the examination of functional traces on stone artifacts. Studies about function indicate specific tasks for which people used the artifacts. Five possible entries include: microwear trace, macrowear trace, residue, no traces of function or not analyzed. While entering data, a user can select different functions inferred from the traces in the attribute 'function_list' and different functional traces from the attribute 'functional_trace_list'; new entries about function can also be added. The number of pieces specifically analyzed for functional traces is entered, as only a sample of the entire assemblage is customarily studied. Also, the quantity of each type of functional trace and the number of lithic artifacts with a given functional trace are entered.

The non-lithic finds are described in the next four tables, as shown in Fig 8. Like the stone artifacts, non-lithic finds generally consist of collections of finds, not single pieces. These tables are each structured in a similar way to facilitate comparison and usually include the find count, specify the material and provide an interpretation.

The table "organic_tools" includes artifacts manufactured from biological materials such as bone, ivory, antler, tooth, mollusk shell, eggshell or wood. Art objects, musical instruments and ornaments are not considered as organic tools, but rather belong in the table "symbolic_artifacts". The raw material of the organic tool is selected from a list, and the number of tools manufactured from each organic material is entered separately. The type of artifact manufactured from the organic material should also be entered.

The table "symbolic_artifacts" includes artifacts that convey an abstract or notional meaning. Symbolic artifacts as understood in ROAD encompass finds that we interpret as evidence of art, music or personal ornamentation. This description consists of the material which bears the symbolism (e.g. stone, bone, shell or ostrich eggshell), category (e.g. art, music or ornament), type (e.g. bead, geometric engraving, human or animal figurine) and type of technology (e.g. drilling, engraving, painting or sculpture).

The table "miscellaneous_finds" includes categories that do not belong to any of the tables described above. Miscellaneous finds include exotic objects that people collected such as pigments (e.g. ochre, specularite or limonite), minerals (e.g. crystals), fossils, bitumen, resin and jet (anthracite coal). The number of each category can be specified. If known, the transport distance of the raw material can be indicated, similar to the table "raw_material".

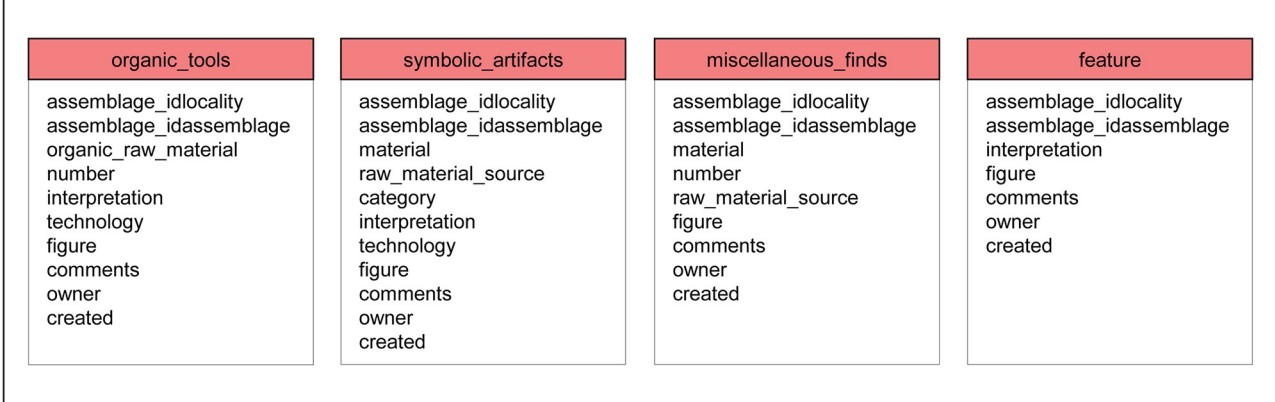

**Fig 8. Tables for entry of non-stone artifacts listing their respective attributes.** Organic tools can be made of bone, tooth, ivory, shell or wood, while symbolic artifacts include art, music and ornaments. Miscellaneous finds such as ochre, minerals or unmodified shell, and features such as burials or stone constructions can also be entered.

The table "feature" describes intangible or conceptual artifacts found at archaeological localities. Features are often not portable and may consist of a hearth, burial, architectural element, soil stain or other elements that cannot be readily recovered as finds. Additionally, our definition of feature includes grouped sets of finds. One example is the Middle Stone Age ocher processing kit described from Blombos Cave in South Africa [27] which consists of an abalone shell containing ochre, charcoal, bone and grinding stones. Other examples are the Gravettian child burials from Krems-Wachtberg, where burials with ornaments are part of a bone and tusk construction and surrounded by ochre in the sediment [28]. While each type of "feature" should be entered separately, individual components of a given "feature" may also appear in the other archaeological tables.

**2.3.2 Detailed assemblage description: Human remains.**   In addition to the archaeological remains described above, the catalog of finds stored in ROAD includes human remains (Fig 9). Descriptions of these finds are stored in the table "humanremains". We define a human remain as a direct and substantial piece of evidence for the presence of humans at a particular locality. In addition to human fossils, this category also includes endocasts, footprints, ancient DNA and other traces resulting from human presence. Human remains may occur in archaeological contexts or in natural archives. Unlike the other categories of finds, human fossils are usually entered as single remains, for example, a complete calvarium, a distal left tibia or a third molar. If the human remains clearly belong to the same individual, for example in the case of a burial, or if they consist of remains that can be attributed to a single individual, the collection

| humanremains |
| --- |
| assemblage_idlocality |
| assemblage_idassemblage |
| idhumanremains |
| accession_number |
| skeletal_element |
| category |
| pathologies |
| anthropogenic_modification |
| figure |
| comments |
| owner |
| created |

| publication_desc_humanremains |
| --- |
| publication_id_source |
| publication_idedition |
| publication_idpublication |
| humanremains_idlocality |
| humanremains_idassemblage |
| humanremains_idhumanremains |
| genus |
| species |
| sex |
| body_mass_min |
| body_mass_max |
| age |
| brain_volume |
| brain_volume_method |
| brain_mass |
| genetic_sample_source |
| roceeh_default |
| comments |
| owner |
| created |

**Fig 9.  Tables for the entry of human remains in ROAD.** Because researchers may interpret human remains differently in different publications, ROAD includes a second table "publication_desc_humanremains", which links a relevant publication to its record in the table "humanremains". The table "publication_desc_humanremains" includes further interpretations about the published description of the human fossils.

of finds is entered as one assemblage. The description of human fossils includes information about skeletal element, side, pathologies and anthropogenic modifications.

A second table, "publication_desc_humanremains", serves as a linking table with hybrid function. It provides further specifications about the fossils, such as genus, species, age, sex, body mass, brain mass and brain volume. Since human fossils may be assigned to different taxa by different researchers, we present each interpretation as a separate entry in this table. Thus, it is possible to include a find with various interpretations. In such cases where researchers are not unanimous about a fossil, the attribute 'roceeh_default' provides our preferred interpretation.

**2.3.3 Detailed assemblage description: Faunal remains.**   Faunal remains include the bones, teeth and antler of large and small mammals, excluding human remains. In addition to the mammals, faunal remains encompass birds, fish, reptiles, amphibians and mollusks. Faunal remains may occur in archaeological contexts or in natural archives. In general, faunal remains are entered as groups of finds and not as single objects.

There are two important reasons why we analyze the faunal remains. The first is to draw conclusions about past human behavior including diet and subsistence based on quantitative and qualitative analysis. For the archaeological analysis of the faunal assemblage, ROAD makes use of the table "animal_remains". This entity includes information about the frequency of skeletal elements and different types of taphonomic modification (e.g. anthropogenic, biogenic or geogenic).

The second is to reconstruct habitat based on taxonomical classification and simple quantitative analysis. For the taxonomical and quantitative analysis of fauna, ROAD makes use of three tables (Fig 10): "paleofauna", "taxonomical_classification" (a look-up table) and "publication_desc_paleofauna" (a linking table with hybrid function). Faunal assemblages entered in ROAD include the taxonomical classification as they are published. Their description consists of both primary and secondary data. Primary data can be classified in quantitative and qualitative terms [29]. The quantitative component includes the number of identified specimens, while the qualitative part covers taxonomic classification. Primary data are stored in the tables "paleofauna" and "taxonomical_classification". Additionally, quantitative secondary data, such as the minimum number of individuals, are stored in the table "publication_desc_paleofauna". Unlike primary data, secondary data need further manipulation or interpretation to gain their meaning. The value of secondary data often depends on the method applied, which means that the same primary data can yield varied interpretations.

**2.3.4 Detailed assemblage description: Plant remains.**   Paleobotanical remains include a variety of plant parts, such as leaves, seeds, wood, pollen and others, or modified remains, such as charcoal. Fossil plant assemblages may occur in archaeological contexts or in natural archives. The analysis of natural paleobotanical assemblages helps to reconstruct former vegetation and past climatic conditions. Archaeobotanical assemblages provide information on human plant use and may help infer different patterns of human land use. Botanical remains are generally entered as groups of finds and not as single objects.

Similar to the faunal remains in ROAD, information about botanical remains is stored in three tables (Fig 11): "plantremains", "plant taxonomy" and "paleoflora". The table "plantremains" characterizes the general fossil flora of an assemblage. Further details about taxonomic composition can be entered in the table "paleoflora", which specifies the plant taxa contained in a fossil flora and provides information about their abundances. The table "plant_taxonomy" lists the systematic details for classifying each plant taxon that occurs in the table "paleoflora".

## 2.4 ROAD resources: Table descriptions and more

We recognize that ROAD is a complex database with a steep learning curve. Therefore, we assembled the "Table Descriptions" (S1 File) as a guide to help users understand the contents

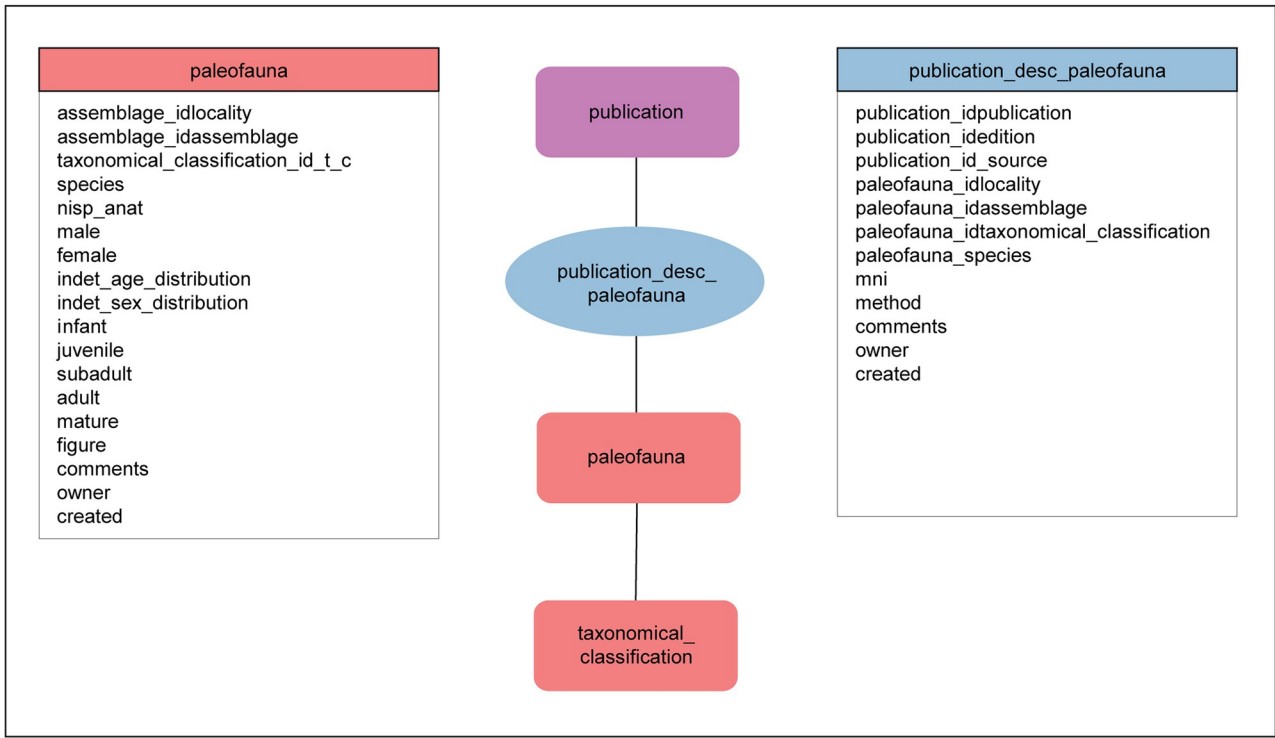

**Fig 10. The table "paleofauna" is linked to the look-up table "taxonomical_classification" and to the table "publication".** The table "paleofauna" is linked to the look-up table "taxonomical_classification". The link joining these tables represents a one-to-many relationship. The table "paleofauna" is also linked to the table "publication". The link joining the tables "paleofauna" and "publication" represents a many-to-many relationship. To describe this relationship, ROAD includes the table "publication_desc_paleofauna" in which every record has a link to the relevant publication, as well as to its record in the table "paleofauna". In addition, "publication_desc_paleofauna" specifies the MNI and the method used to calculate it because MNIs for faunal remains can differ depending on the publication and method applied.

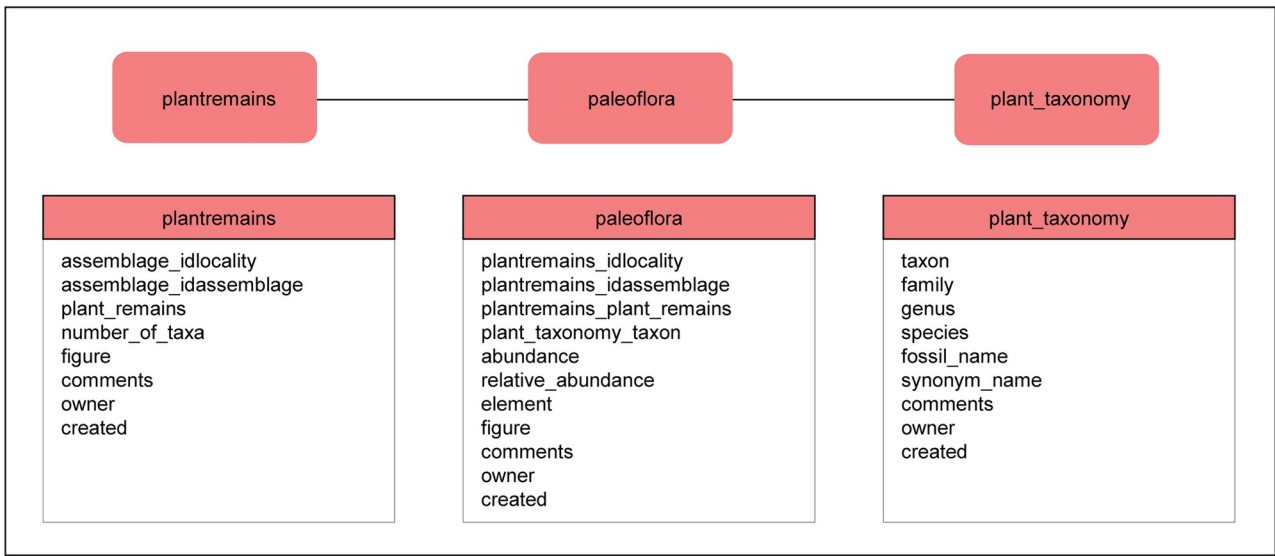

**Fig 11. Tables for botanical finds and their relationships.** The link between "plantremains" and "paleoflora" as well as the link between "paleoflora" and "plant_taxonomy" each represent a one-to-many relationship.

of the database. The guide is accessible in ROAD as a downloadable PDF under "ROAD resources" and includes a description of the aims and objectives of the ROCEEH research center and an overview of ROAD, with tips for users. It focuses on describing each table and provides examples of data entry. Furthermore, it includes a comprehensive description of each attribute with instructions, guidelines, definitions and specifications for data entry. The information contained in the guide is also accessible through online icons incorporated into every table view and each data entry form. These icons allow immediate access to information about a specific table or attribute when the user is unsure of its purpose or meaning.

A further ROAD resource is the "ROAD database model", or Entity Relationship Diagram (S1 Fig) which maps the detailed structure of the database as compared to the schematized version presented in Fig 1. Another useful resource is the "ROAD Manual" (S2 File) which provides general information about the database and describes how to write queries. The manual also illustrates how to use the Map Module to visualize data in ROADWeb, including the ability to plot the results of predefined searches and user-defined queries onto an interactive map with different background layers.

## 3. The ROAD database application—ROADWeb

Here we discuss some of the functional aspects of the database, as it incorporates an entire suite of applications made available through ROADWeb. First we describe technical issues surrounding the implementation of the database and the functions available to each class of user. Then we illustrate what ROAD offers by delving into issues of handling data including entry, editing, exploration, querying, visualization, import and export.

### 3.1 ROADWeb implementation

The backend of ROADWeb uses a PostgreSQL database management system, which is freely available and open-source, emphasizing extensibility and SQL compliance. The web-based frontend, also known as the graphical user interface, uses scripts written in the programming languages PHP, JavaScript and HTML. A user can easily access the frontend through a browser, which allows the user to communicate with the database in order to: a) enter and manage data; b) query, analyze and visualize data; or c) import and export data. As a rule, we test ROAD using Firefox. We note that other browsers such as Chrome, Edge and Safari are also compatible.

### 3.2 ROAD user groups

We divide ROAD users into four groups (Table 1) based on their level of access: 1) the general public with interest in early human studies and scholars who want to retrieve basic archaeological information without logging in; 2) associated scientists involved in the analysis and visualization of ROAD data and geographic analysis; 3) associated members involved in data

**Table 1. List of ROAD user groups and their capabilities.** Group 1 can access ROAD without a login, while groups 2–4 require a user id and password to log in.

| Group | ROAD Users | Login | Capabilities |
|---|---|---|---|
| 1 | General public | χ | Visualization of data, PDF generation, simple query |
| 2 | Associated scientists | √ | All the above + individualized querying |
| 3 | Associated members | √ | All the above + selected data entry |
| 4 | ROCEEH team | √ | All the above + full data entry, quality control, development of tools |

exchange, input and analysis; and 4) ROCEEH team members with full access to ROAD, including development, programming and quality control of the data entered.

Group 1 has the ability to visualize data, generate a ROAD Summary Data Sheet for a selected site and conduct basic queries about the "what, where and when" in a simple, user-friendly format without the need to log into ROAD. Group 2 has the additional ability to write and execute queries, map the results and download data. Group 3 has the further benefit of being able to enter data into selected tables and edit their own data, but not other datasets. Finally, group 4 has full access to all aspects of ROAD, such as entering and editing all data, implementing quality control and developing other tools.

Quality control is achieved through various means. The most important tool is the ROAD Summary Data Sheet, which provides an overview of a given locality. With this PDF we check whether data are correctly linked and note which data were not yet entered. In addition, we perform routine controls of data using the table views, queries and a function called 'data control'. With these, we check for duplicates and inconsistencies in data entry, as well as missing data.

Since group 1 can access ROAD without a user account or password, this level is open to everyone (www.roceeh.uni-tuebingen.de/roadweb). Anyone interested in advanced studies using ROAD can apply for an account. Thus, access to groups 2–4 requires authorization, which means that a user signs and returns a "User Agreement" (S3 File) to set up an account with a password. The agreement states that a user agrees to use the data in accordance with generally accepted academic standards. In particular, this means that users of ROAD should cite the original publications and acknowledge ROAD as the source of data. However, when large amounts of data with many references are retrieved, it may be more expedient to cite ROAD as the source. The access date of a ROAD query should always be stated, as the database grows continuously. ROAD currently offers advanced services to more than 170 authorized users.

## 3.3 Data entry and editing

Only users of groups 3 and 4 are allowed to enter and edit data in ROAD. While members of group 3 are restricted to editing the datasets they personally entered, those of group 4 are permitted to edit any dataset. Data entry is steered through structured workflows, one for each of the four scientific disciplines: archaeology, paleoanthropology, paleofauna and paleobotany. An additional workflow is reserved for entering bibliographic information.

Each workflow guides a user through the tables necessary to complete for the specific discipline selected. Since the workflows are hierarchical, they should be completed in sequence. The four scientific workflows share common tables, including "locality", "geological layer", "geological stratigraphy" and "assemblage". However, archaeology makes use of additional tables for "archaeological layer" and "archaeological stratigraphy", as these tables make it possible to ascribe a culture to a layer.

While the input of bibliographic information also follows a workflow, we found that it was simpler to first enter the sources into a reference management program like EndNote and then export new references as a BibTeX file. We then import the BibTeX file using ROADWeb's import function, which transfers the data into the appropriate tables and attributes. The user then links the publication title and its associated bibliographic information to the various tables beginning with "publication_desc".

## 3.4 Analytical tools and visualization

Anyone can access ROAD without a password in the following two ways: 1) either directly through the ROAD website (www.roceeh.uni-tuebingen.de/roadweb), or 2) through the Web-GIS Map Module of ROADWeb (www.roceeh.uni-tuebingen.de/roadweb/map_modul). Both

pathways offer a user the ability to search for information in ROAD based on fixed variables such as locality name, region, assemblage type and age (Fig 12). A user can further specify an interval of time numerically or by selecting a predefined period from the global chronostratigraphical correlation table based on either series, substage, marine isotope stage or paleomagnetic chron and subchron. The search results are plotted on a map showing the spatial distribution of the localities that meet the specified search conditions.

Additionally the ROAD Map Module allows searches in two external databases: 1) Neotoma Paleoecology Database and 2) Neogene Quaternary Mammals Database (NQMDB). Neotoma contains paleoecological data from the Pliocene and Quaternary including plant remains, mammals and mollusks. Neotoma has a strong focus on the Americas and can be accessed without a password. NQMDB hosts lists of both macro and micromammals for Pleistocene fossil localities in western Eurasia, but can only be accessed by obtaining a separate login from its creators at the *Centro Nacional de Investigación sobre la Evolución Humana* (CENIEH) in Burgos, Spain. ROCEEH provides maps of vegetation, biome, hydrology and bathymetry among a large catalog of cartographic resources. These capacities show ROAD's potential to link with further external resources, and vice versa.

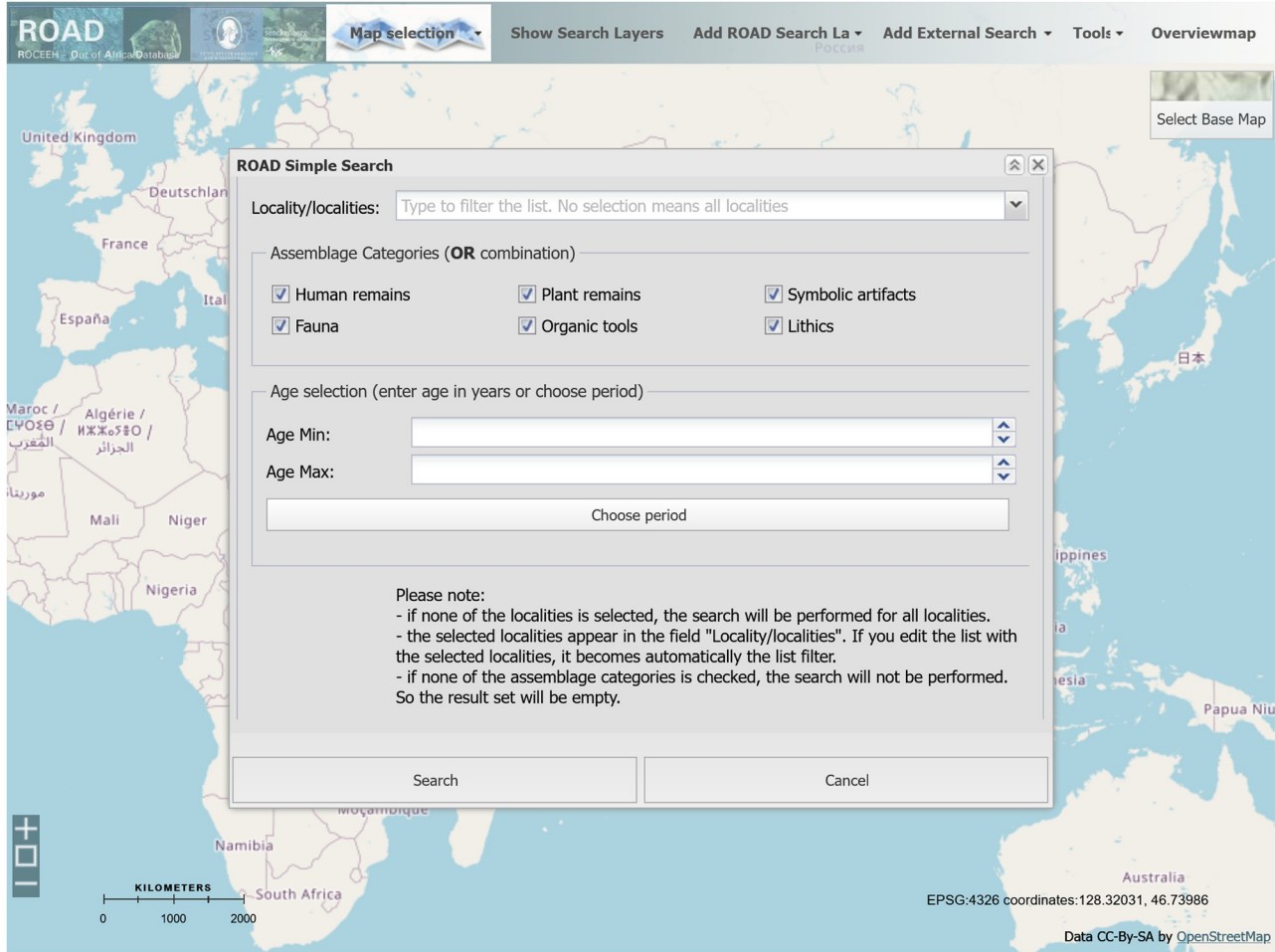

**Fig 12. The ROAD map module.** The Map Module (background) and a pop-up window (foreground) used to conduct a simple search in ROAD. A simple search can be executed based on questions about the where (localities), what (assemblages) and when (age). Map credit: OpenStreetMap.

Authorized users (groups 2–4) can search and analyze data using the "SQL query tool" and the "time slice tool" to generate quantitative analytical results. With the SQL query tool, an authorized user with no knowledge of programming can graphically compose a query (Fig 13). A query joins together several tables and can specify conditions to produce a list of results meeting those specifications. After a query is performed, a user can export the data as Comma Separated Values (CSV) or visualize the results using the Map Module. The time slice tool performs binning to smooth data over specified intervals in a time series analysis. The tool generates a table listing the number of localities and the number of assemblages meeting the criteria of the selected query. With the time slice tool a user can export the results as a CSV file and display the results graphically. The benefit of such a tool is to visualize time series, as used in a study of how the use of ochre changed over time in Africa [30].

Other features of ROAD allow a user to display information about a selected locality including its location on a map and its geological and archaeological profiles (Fig 14). The interactive geological profile includes information about each layer, such as its name, sedimentology, thickness, and most importantly, age range. By clicking a geological layer, a user can also

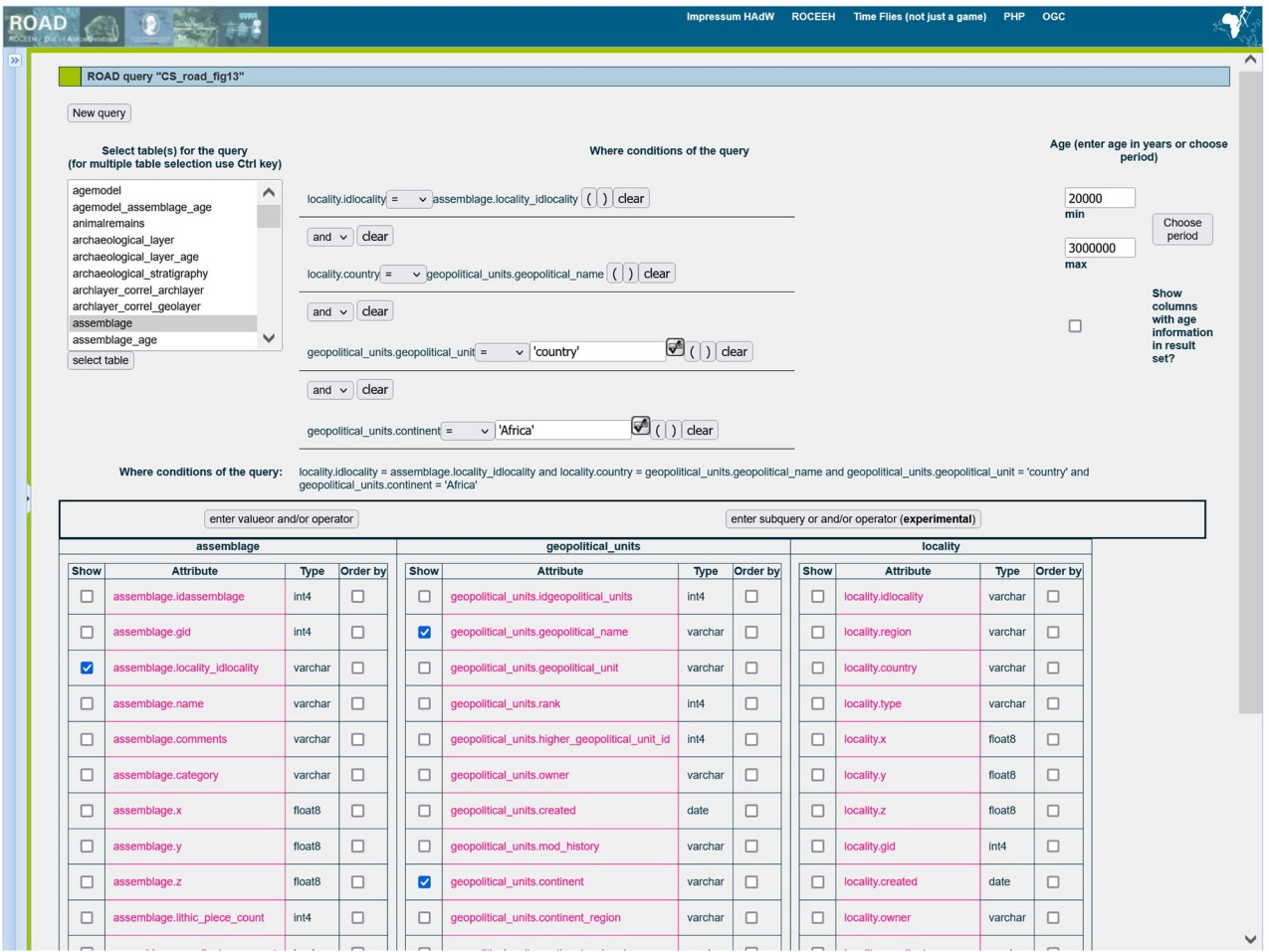

**Fig 13. View of the SQL query tool for composing queries.** While knowledge of programming is not required to use ROAD, it is necessary to learn how to use the interface itself. This can be accomplished with help from the ROAD Manual (S2 File) and other supporting documentation.

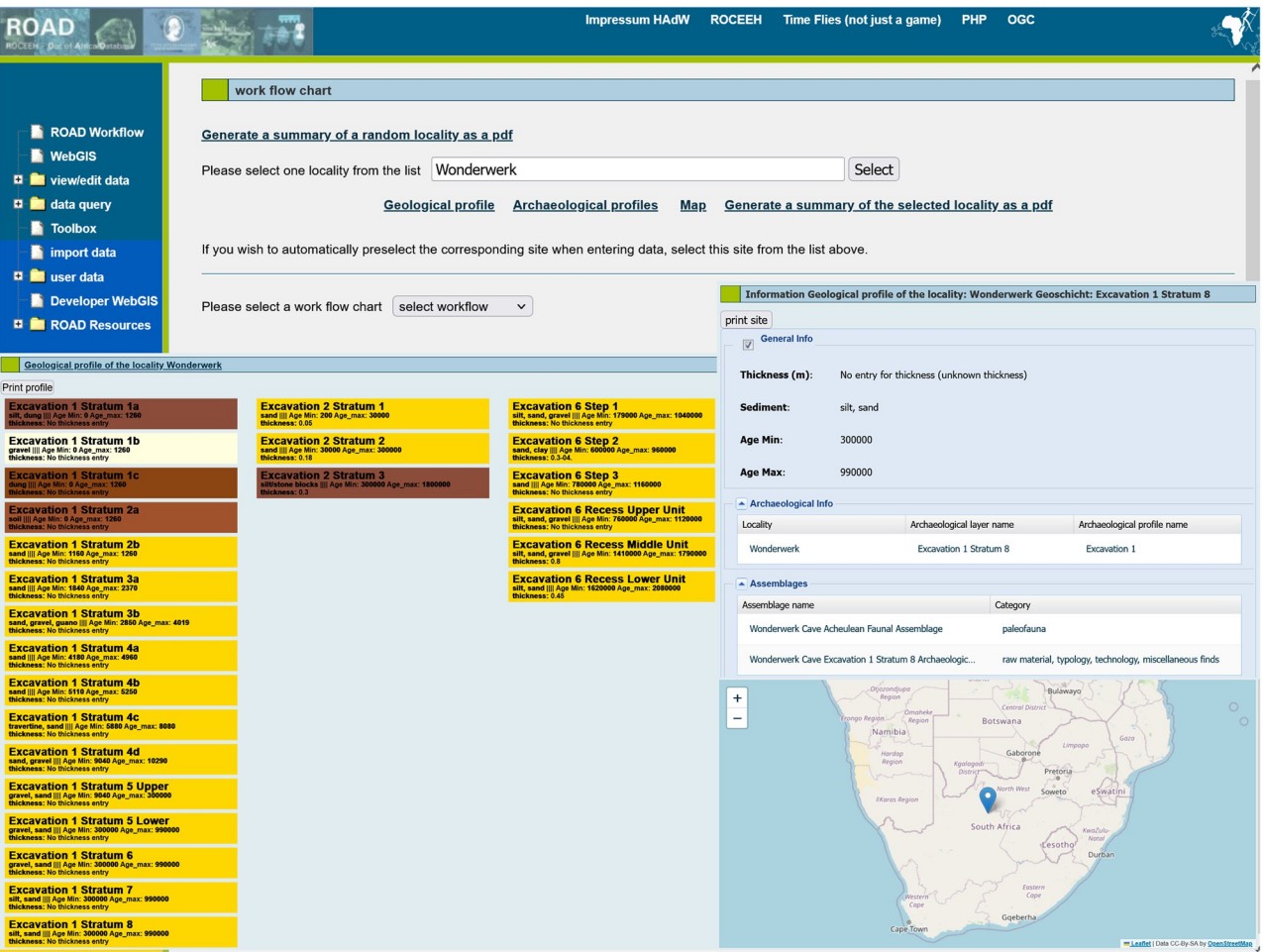

**Fig 14. Examples of ROAD search possibilities.** Examples from the locality Wonderwerk Cave in South Africa. A view of interactive geological profiles (left), associated archaeological layers and assemblages in a selected geological layer (upper right) and map view (lower right). Map credit: OpenStreetMap.

display the archaeological layers and assemblages associated with it. An archaeological profile includes the cultural attribution of its layers.

## 3.5 Data export

To make the information stored in ROAD more accessible, ROAD provides several formats to export data: 1) site summary data sheets as PDF documents; 2) data export as a CSV; 3) creation of an Application Programming Interface (API) on demand; and 4) Resource Description Framework (RDF) export for SPARQL querying.

**3.5.1 Site summary data sheets.** A site summary data sheet provides a user with an overview of a locality entered in ROAD, whether they know the structure of ROAD or not [31]. As the name suggests, the data sheets include a statement summarizing the site as well as selected data about its location, geological and archaeological profiles, layers, ages, assemblages and references (S4 File). While these sheets are useful for gaining an overview of a selected locality, they do not contain all information stored in ROAD; nor can they bring together information about more than one locality.

**3.5.2 Data export as CSV.** To gather further information from ROAD, an authorized user can export data as a table in CSV format. This can be accomplished from a single table with the help of filters (e.g. all localities in France, all caves, or even all localities in France that are caves). But filters cannot search for combinations of data, such as France and Germany, or caves and open-air sites. In such cases, or if information from more than one table is required, it is necessary to write and perform a query using the SQL query tool (e.g. all localities with human remains from caves and open-air sites in Germany and France containing stone artifacts and equids).

**3.5.3 Creation of an API.** With help from the ROAD team, collaborating scientists can obtain an API to generate data directly from ROAD. Thus, ROCEEH helps scientists formulate their SQL queries, which they can then run in their own browser. One such example is a query for African localities with either lithics, hominins or paleofauna dated between 120,000 and 1.8 million years ago, aggregated at the level of locality: https://www.roceeh.uni-tuebingen.de/api/Africa-localities-lithics-hominins-paleofauna-120ka-1800ka.php.

**3.5.4 RDF export for SPARQL querying.** To incorporate ROAD into the Semantic Web, data need to be saved in the form of RDF triples. For this reason, we created an ontology of ROAD to map out all of its tables and attributes. We then defined an RDF data export based on this ontology. While the export is still in development, a beta version of the RDF export of ROAD data already exists in ROADWeb. With this feature, it is possible to query the RDF data using SPARQL querying language. We are currently expanding this functionality, as it represents an essential component to prepare ROAD for the future with Linked Open Data. This improvement will increase the sustainability of the database long after the ROCEEH research center ends in 2027.

In fact, the beta version of the RDF export is fully functional and accessible through a SPARQL endpoint. The RDF export is updated every six months and serves several purposes: as a backup, as the basis for various analyses, and for retrieving data for use in maps and other resources, which also serves to increase the visibility of ROAD. Consequently, ROAD can be queried through the Semantic Web and linked to other entities in it. Where practical, we incorporate external ontologies (e.g. WGS84 for geodata, Dublin core for metadata and DBpedia with its wide range of information). While ROAD contributes to the development of community standards, the lack of established standards in archaeology hinders the interoperability of ROAD in its current state. The establishment of such community standards is a task that must be addressed by the entire discipline.

## 4. Results

The results presented in the following text and figures provide an overview of ROAD based on database queries conducted on 21 February 2023. We examine details and statistics about: 1) the spatial and chronostratigraphic coverage of the database; 2) its degree of interdisciplinarity; 3) the nature of its publication sources; and 4) the geographical distribution of cultural entities. We also show ways in which ROAD is being employed by the ROCEEH team and other collaborating projects, and then discuss future perspectives for the database after the research center comes to a close.

### 4.1 Spatial and chronological coverage

In geographic terms, localities in ROAD are almost evenly divided among the continents of Europe (n = 612), Africa (n = 600) and Asia (n = 512; Fig 15A and 15B). Nonetheless, we note differences in the spatial distribution of the assemblages, with the greatest density of localities and assemblages occurring in Europe and along Africa's Rift Valley and in South Africa.

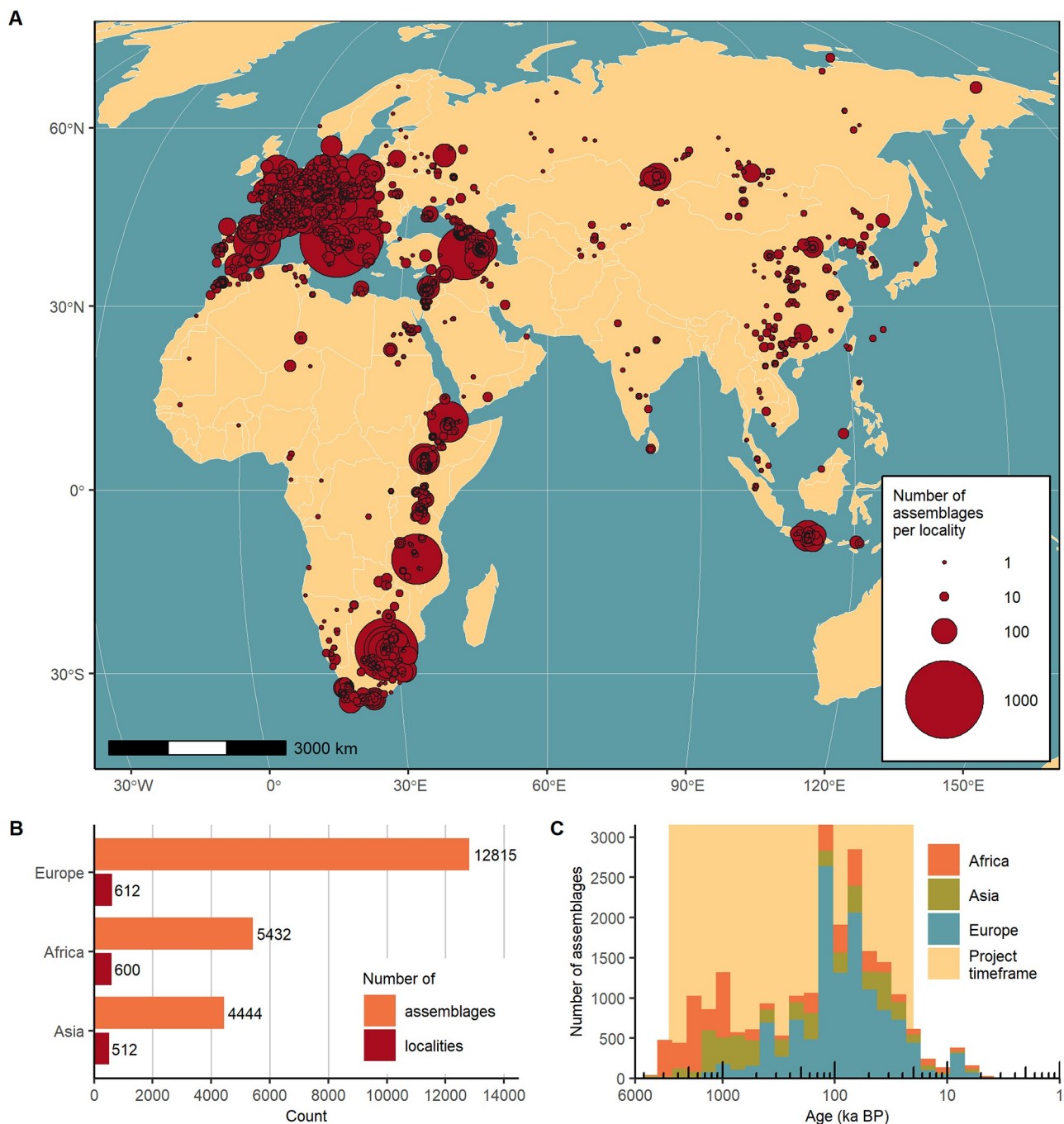

**Fig 15. Spatial and chronological coverage of assemblages contained in ROAD as of 21 February 2023.** The assemblages include archaeological finds and human, faunal and plant remains. (A) Spatial distribution of localities in Africa and Eurasia showing the richness of the assemblages. (B) Number of localities and assemblages per continent. (C) Temporal distribution of assemblages based on their mean ages and graphed as stacked columns. Due to the log-scale of the x-axis, the time contained within each bar increases with age. Map credit: Made with Natural Earth (public domain).

**Table 2. List of the number of localities and assemblages entered in ROAD.**

| Discipline | Localities | Assemblages | Mean number of assemblages per locality |
|---|---|---|---|
| Archaeology | 1198 | 5107 | 4.3 |
| Paleoanthropology | 447 | 5340 | 11.9 |
| Paleontology | 642 | 3106 | 4.8 |
| Paleobotany | 285 | 9308 | 32.7 |
| TOTAL with assemblages | 1724 | 22,691 | 13.2 |
| Sites entered, no assemblages | 177 | — | — |
| Sites reviewed, but not entered | 476 | — | — |
| TOTAL localities | 2377 | — | — |

In addition, we show the number of localities whose assemblages have not yet been entered (i.e. those still in progress), and those localities we reviewed, but chose not to enter (e.g. too young, no dating, publications not accessible). Since localities and their assemblages can appear in one or more disciplines, the total number of localities with assemblages does not add up.

There are more paleoanthropological and paleobotanical assemblages per locality compared to archaeology and paleontology because of the way these are entered. Human remains typically represent one skeletal element, except when they belong to the same individual; natural pollen archives usually are drillings or outcrops sampled at a high temporal resolution, resulting in a high number of assemblages per locality.

Table 2 shows the breakdown of localities and assemblages by discipline. In chronological terms, a small part of the data span beyond the project timeframe of three million to 20,000 years before present, while the largest portion of the record dates between roughly 120,000 and 50,000 years (Fig 15C).

In 2008 and 2009 the team met regularly to discuss content and define terminology. During this time, we conceived and implemented the database structure and started entering data in October, 2009. While the project staff coordinated the scientific aspects of data entry, student research assistants reviewed the literature and entered the majority of data. In addition to students with good knowledge of English, French and German, we also sought languages beyond this skill set. In this way, we gained access to information published in Spanish, Italian, Russian and Chinese, among other languages. We conferred with international colleagues to select localities for data entry, and often relied on them to obtain obscure publications, gain clarifications, make corrections, and occasionally enter their unpublished data. In addition, we conducted periodic rounds of quality control to minimize discrepancies in data entry including technical errors.

To prioritize which localities to enter, we chose well-published sites accessible in languages the team could master. We preferred sites with clear stratigraphy and reliable dating results. Within a given country, we selected representative sites that reflect the local knowledge base and strived to enter a sample of periods within ROCEEH's timeframe. Of the localities entered, there are 982 open-air sites, 615 cave sites and 187 rockshelters, in addition to 131 outcrops, 61 drill cores and 58 quarries. For 476 of the localities reviewed (Table 2), we elected not to enter assemblages for a variety of reasons including inaccessible publications or those of questionable scientific value, unclear stratigraphy, lack of conclusive dating, and sites younger than the project timeframe. Should reliable information about a locality become available in the future, we can complete its data entry.

In addition to the geocoordinates that mark the location of each site, 12,286 individual datasets position the assemblages temporally. These data come from the three "age" tables, "geological_layer_age", "archaeological_layer_age" and "assemblage_age". Of these dates, 10,720 comprise the most commonly applied methods of absolute dating in ROAD (Fig 16). In order of frequency, these are radiocarbon ($^{14}$C; 47%), optically stimulated thermoluminescence

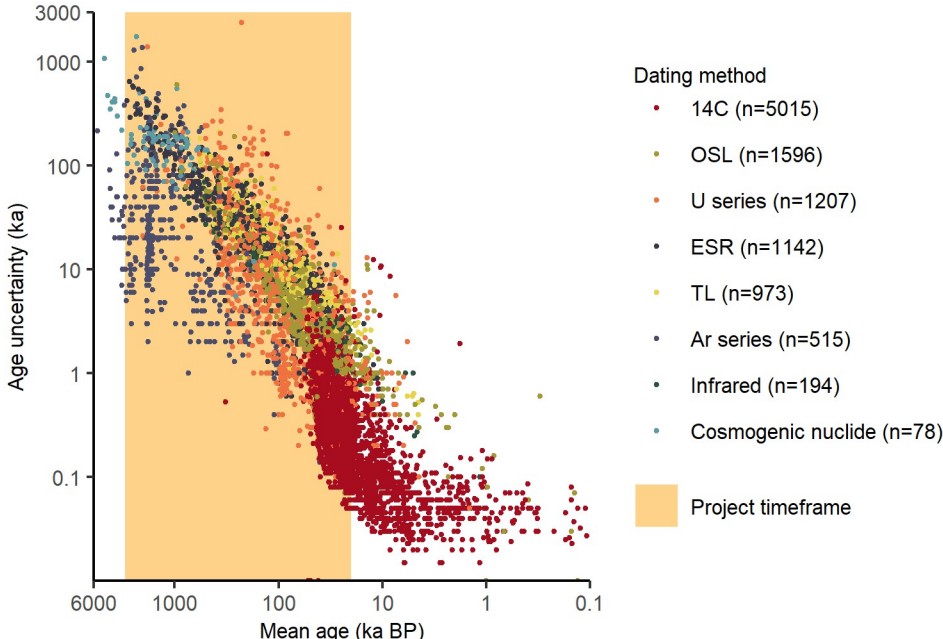

**Fig 16. Absolute dating methods, mean ages and age uncertainty.** We include the most common dating methods (n = 10,720) entered in ROAD, but exclude mixed methods and less common absolute dating methods (n = 1566) as well as relative ages. When the age uncertainty on the y-axis (i.e. the difference in positive and negative standard deviation) is asymmetric, we report the higher uncertainty. (See text for explanation of dating methods.).

(OSL; 15%), uranium-series dating (U/Th, U/U, U/Pb; 11%), electron spin resonance (ESR; 11%), thermoluminescence based methods (TL, ITL; 9.1), argon dating (Ar/Ar, K/Ar; 4.8%), infrared methods such as stimulated luminescence and radiofluorescence (IRSL, IRSL post-IR, IR-RF; 1.8%) and cosmogenic nuclide dating (Al/Be; 0.7%). We excluded mixed or pooled dating methods (e.g. coupled ESR and U-series) and those that occur less frequently in ROAD (e.g. rubidium-strontium or fission track). In all we excluded 1566 (13%) dates from the analysis shown in Fig 16. Additional information about age comes from methods of relative dating and is reported in the table "geological_stratigraphy". Relative methods include correlation by geological or cultural stratigraphy, as well as such diverse sources as magnetostratigraphy, tephrostratigraphy, biostratigraphy, marine isotope stage, amino acid racemization and obsidian hydration dating.

## 4.2 Interdisciplinarity

ROAD structures information from four main disciplines: archaeology, paleoanthropology, paleontology and paleobotany. Assemblages are placed within geological and archaeological layers which can be correlated to each other through linking tables. To obtain an overall picture of the interdisciplinary nature of ROAD, we can relate the signals from each discipline with one another. In total, 1724 localities include at least one assemblage associated with the four disciplines (Fig 17). Cultural remains are present at 1198 localities, human remains at 447 localities, faunal remains at 642 localities and plant remains at 285 localities. One-third (589: 34%) of the localities provide information on more than one discipline. This can be broken down even further into the number of localities that combine two (360: 21%), three (199: 12%) and all four disciplines (30: 2%).

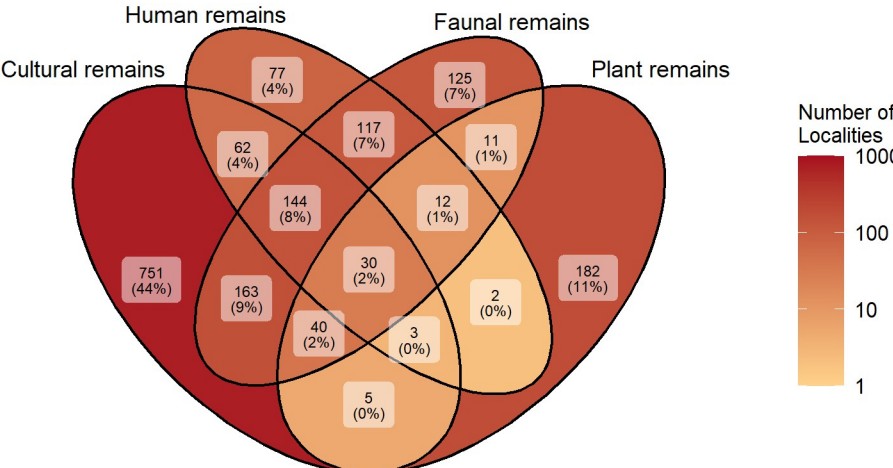

**Fig 17. ROAD's database structure enables interdisciplinary work.** The Venn diagram visualizes how many localities are associated with the four main categories (archaeological, human, faunal and plant remains) and how they intersect. The darker the color, the greater the number of localities. Note that the sizes of the areas are not represented proportionally. Percentages are related to the total number of 1724 localities associated with at least one assemblage category.

The paleoanthropological data profit most from the database's interdisciplinary approach, as 83% (370) of all localities with human remains complement information from one of the other disciplines. The value for faunal remains is similarly high, with 81% (517). While 447 localities with cultural remains are enriched with content from other subjects, about two-thirds of the archeological sites (751: 63%) are not associated with data from the other disciplines. This trend likely results from our focus on archaeological sites, many of which did not undergo multidisciplinary investigations in the past. It could also result from taphonomy, simply because stone artifacts are generally more durable than organic materials. Taphonomic processes likely account for the comparably low number of localities with plant remains, especially in Africa where organic preservation is low. Furthermore, their lower degree of overlap (103: 36%) with the other disciplines may relate to their deposition in diverse natural environments. For example, plant remains are usually found in lake or peat sediments, where humans and animals are typically not present or preserved.

## 4.3 Sources of publication

ROAD contains 5020 titles, with the oldest published in 1865 (Fig 18A). Despite this long history, the median publication year is 2004, and 40% of titles were published since ROCEEH began in 2008. Even though newer literature predominates, many titles were published before the quantitative revolution, let alone the concept of open data. Thus, ROAD provides access to structured and machine-readable legacy data, which represents a substantial share of the dataset.

The curve in Fig 18B shows that the literature sources in ROAD are quite disparate. On one hand, a large concentration of titles are published in a small number of international journals. In fact, 29% (n = 1476) of the publications in ROAD are sourced from just ten journals, each contributing between 56 (from PLoS One) and 370 (from Journal of Human Evolution) titles. On the other hand, 1028 of the titles are unique, coming from books, theses and reports, accounting for 20% of the literature. Between these two extremes, book series, conference

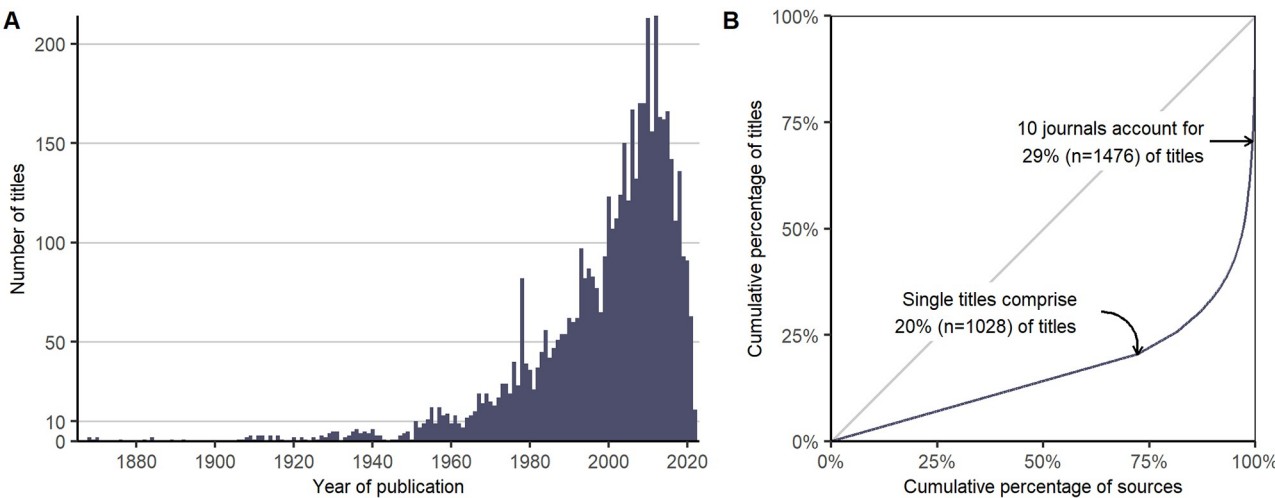

**Fig 18. Data sources used in ROAD.** ROAD is sourced from 5020 publications spanning the last 158 years. (A) Distribution plot of the years of publication. (B) Lorenz curve showing the relative concentration of sources. The y-axis shows the cumulative percentage of titles, while the x-axis shows the cumulative percentage of 1424 sources. A source can be a single book, thesis or report, but also a journal with hundreds of titles (i.e. articles). The further the Lorenz curve deviates from the dashed line of equality, the more disparate or concentrated the distribution.

reports, small journals and journals with regional foci published in various languages complete the picture. In summary, the publication record, as represented in ROAD, is concentrated in a few large international journals published mainly in English, which have, at least historically, been dominated by Western authors from comparatively well-funded research institutions. Nonetheless, a substantial part of the knowledge is published in smaller circulation and in many languages. Thus, ROAD is particularly valuable because it offers information beyond Western centers of research.

### 4.4 Cultural periods

Another way to view the content of ROAD is to examine the cultural periods which are stored as 154 separate entries in the table "archaeological_stratigraphy". The cultural periods are defined through chronological as well as geographical ranges. Based on determinations published by the authors at a given locality, we ascribe a cultural period to each archaeological layer. In many cases, the cultural period can only be defined in broad terms, for example, Early, Middle or Later Stone Ages and Lower, Middle or Upper Paleolithic. In Fig 19 we observe that localities with archaeological assemblages from the Upper Paleolithic/Later Stone Age and Middle Paleolithic/Middle Stone Age are more plentiful, while the Lower Paleolithic/Early Stone Age occurs less frequently. In terms of technocomplexes, the Mousterian and Acheulean are most common, but such attributions usually do not help to better define the age of those assemblages. On the other hand, when chronologically limited technocomplexes such as Aterian, Howiesons Poort or Aurignacian are present, the age range of the technocomplexes helps refine the period of occupation (Fig 20).

### 4.5 Research based on ROAD data

Several ongoing and completed studies make use of the data stored in ROAD. Some of the investigations were conducted by the ROCEEH team, while others constitute joint efforts with external researchers, or result from independent research. In all cases, ROCEEH provides the

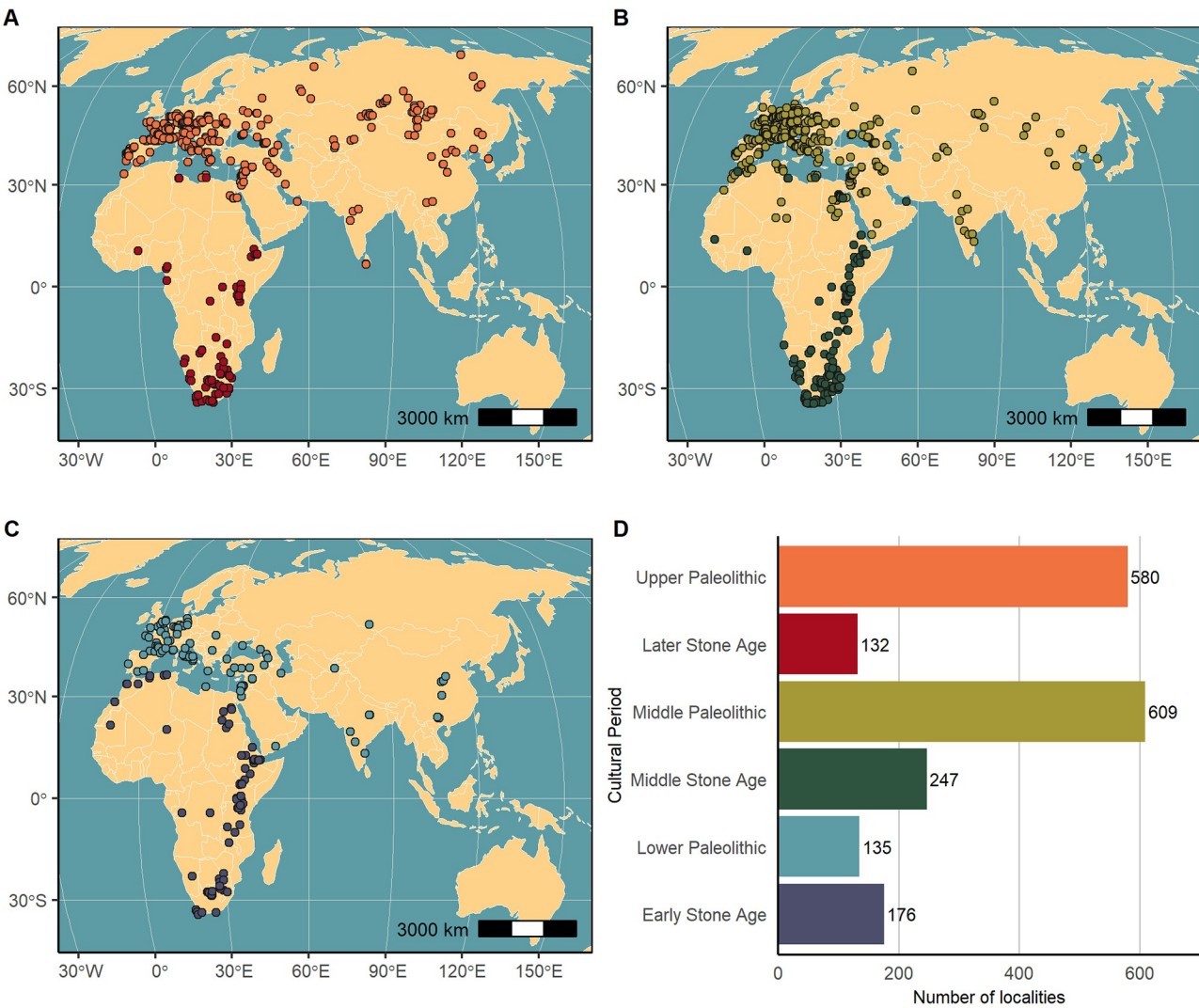

**Fig 19. Chronological distribution of archaeological assemblages.** ROAD contains Paleolithic assemblages attributed to six broadly defined cultural periods. Distributions of (A) Upper Paleolithic and Later Stone Age localities, (B) Middle Paleolithic and Middle Stone Age localities, and (C) Lower Paleolithic and Early Stone Age. (D) The frequency of the main cultural periods by locality, with colors coded to inset maps (A), (B) and (C). Map credit: Made with Natural Earth (public domain).

data from ROAD without constraints, with the expectation that authors cite data from ROAD as described in the User Agreement (S3 File). While ROCEEH readily provides access to data for the entire community, we recognize that not all of our collaborative efforts lead to publications. Rather, the data provide researchers a basis to explore new ideas about specific research questions.

Table 3 provides an overview of some of ROCEEH'S collaborative efforts. Previous studies focused on Africa simply because these data were entered first. Thanks to expanded data entry in Europe and Asia, the latest studies examine larger extents of Africa and Eurasia. In the following sections, we summarize highlights from some of the published studies that used data obtained from ROAD.

Kandel and colleagues [32] discuss how behavioral flexibility allowed Middle Stone Age people in southern Africa to cope with changing situations. Using an approach developed by

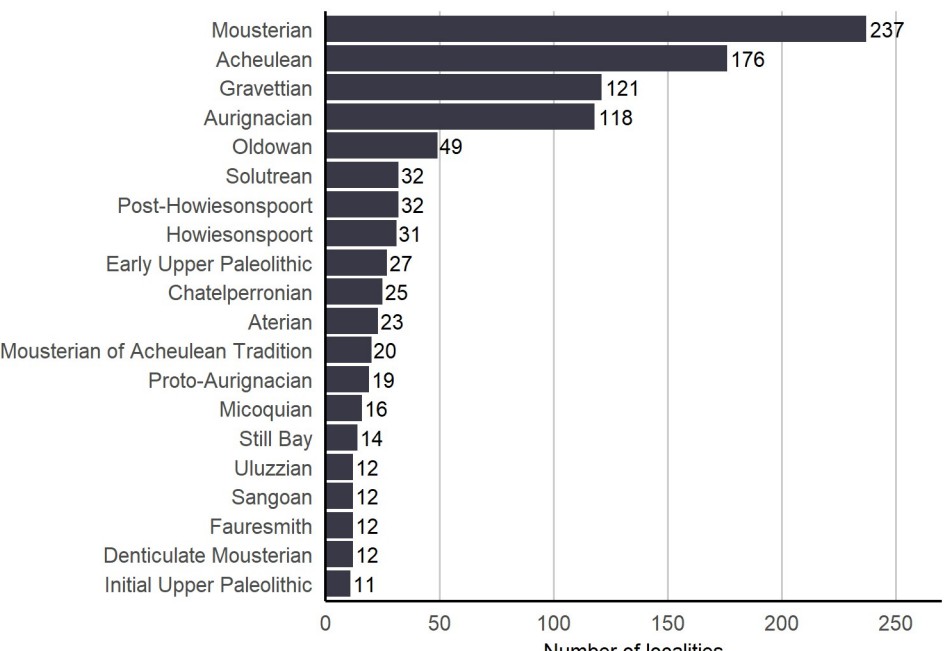

**Fig 20. Distribution of some technocomplexes in ROAD.** The bar chart shows some of the most frequent cultural entities (technocomplexes) contained in ROAD.

Haidle et al. [33], the authors base their analysis on the eight-grade model for the Evolution and Expansion of Cultural Capacities. They conclude that the flexibility of humans to adapt to changing climatic and environmental conditions functioned as a driver of cultural evolution.

With ROAD as his data source, Archer [34] studied the production of stone tools at Late Pleistocene archaeological sites in Africa, modeling the evidence from different artifact types against estimates of population density over time. The results of his study suggest that climate change and demography likely interact, as shown in the ways early modern humans expressed their culture in Africa during the Late Pleistocene.

Sommer et al. [35] conducted an experimental study by analyzing the archaeological data stored in ROAD using methods gained from social network analysis and adapted from big data cartography. This macroscale approach examines similarities in material culture on a continental scale covering a time range of several hundreds of thousands of years. The results

**Table 3. List of studies making use of ROAD data.** This table shows completed and ongoing research supported by ROAD, in chronological order.

| Subject | Geographic focus | Group | Status |
|---|---|---|---|
| MSA behavioral flexibility | Southern Africa | ROCEEH | Published [32] |
| Stone tool technology at Late Pleistocene sites | Africa | MPI-Leipzig | Published [34] |
| Cultural similarity of technocomplexes | Africa | ROCEEH | Published [35] |
| Development of ochre use | Africa | ROCEEH | Published [30] |
| Environmental niche of African MSA | Africa | Jena + ROCEEH | Manuscript prepared |
| Population structure & paleodemography | Africa + Eurasia | Cologne + ROCEEH | Study ongoing, but see [36] |
| Environmental niche of Neanderthals | Eurasia | Cambridge + ROCEEH | Study ongoing |
| Cultural transition from Middle to Upper Paleolithic | Europe | Siena + ROCEEH | Study ongoing |

illustrate the spatial characteristics of cultural taxonomies, but also their shortcomings and biases, as well as the incompleteness of the archaeological record.

Dapschauskas et al. [30] used data in ROAD to investigate the increase in ochre use over time during the African Middle Stone Age. The authors observed three steps in ochre use, which they termed the initial, emergent and habitual phases. Not only did the number of localities increase from phase to phase, but also the amount of ochre, and the percentage of localities yielding ochre. The authors interpret this as a sign that ochre was instrumental for symbolic communication among early modern humans and facilitated their expansion beyond Africa.

### 4.6 Tracking progress

Although each disciplinary team followed a slightly different strategy, our overall approach examined Africa first, after which we switched focus to Europe and Asia (Fig 21). We started entering archaeological data in southern Africa and then covered eastern, central, northern and western Africa over the next six years. After a brief period of entering data from the Levant, we transferred our attention to Europe during the following six years. With the addition of two Russian and two Chinese speakers to our team, we expanded data entry into Asia covering the former Soviet republics and China. By the beginning of 2023, there remain just a handful of places where data entry is not yet complete. These regions include the Iberian Peninsula, Eastern Europe, the Levant, Southeast Asia and Western Africa. Work to fill the remaining gaps continues, and we expect to complete coverage of Africa and Eurasia before the research center draws to a close. Of course new sites—and updates to old sites—continue to appear in the literature, and we strive to incorporate such data into ROAD.

## 5. Discussion

In the first parts of this paper we discussed the project background and covered technical aspects related to creating the large-scale research database, ROAD. Then we examined its

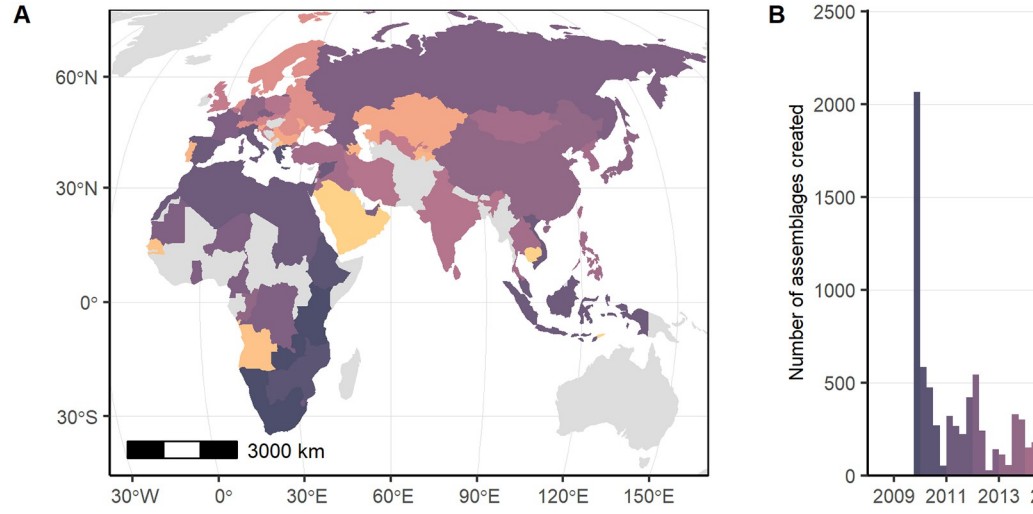
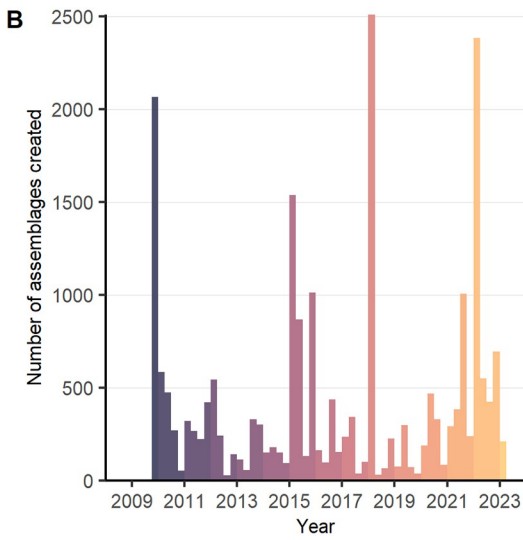

**Fig 21. Development of the ROAD database over time.** (A) Map showing the year when an assemblage was first created within a country. Color codes correspond to the timeline on the right, with gray countries not yet containing sites. (B) Number of assemblages created per quarter year over the lifespan of ROCEEH. Map credit: Made with Natural Earth (public domain).

contents and provided an overview of its structure, basic functionalities and applications. In the discussion we recap some of the lessons learned, look at the utility of large-scale research databases and address aspects that look towards the future of open data, and with it, the future of ROAD itself.

### 5.1 Standardization and openness in human evolutionary studies

Above all, our experience with ROAD shows the importance of creating a standardized framework for data collection. In order to facilitate the large-scale entry of data, we needed to select common denominators, establish hierarchies, define tables and attributes, set up guidelines for data entry and create vocabularies and definitions to assure a high degree of reproducibility. Furthermore, we designed tools to perform quality control of the entered data. These factors were necessary to achieve a high degree of confidence in the dataset.

We recognize that aggregating such a large-scale database requires a degree of openness in human evolutionary studies. Data available as pdfs through online publications including, books, journals and theses facilitated the entry of data and allowed us to compile a digital bibliography with the click of a button on the publisher's website. Having digital data also expedited our search for specific pieces of information. More time-consuming was the customary access of literature made available through libraries. Finally, we are thankful that many researchers gladly shared their articles with us, often discussing their opinions and providing data freely.

### 5.2 GIS implementation and spatial analysis

Our implementation of ROAD as an interactive geographic information system brings certain advantages. First, the geographical information serves as a key for merging interdisciplinary data [37]. Observations in ROAD can be pinned to a specific place and time, which allows us to understand their spatio-temporal relationship and append various disciplinary details as attributes. Second, it enables us to study human evolution through a geographic perspective by examining the expansion of early humans through different lenses, for example: the shape of landmasses [38], climatic effects on human habitats [39, 40] and human bodies [41], or the susceptibility of expansion routes [42–44]. New insights can be gained when ROAD's point data, which cover the where, what and when, are integrated with the growing amount of spatially and temporally continuous datasets about the past [45, 46], including climate models [47, 48], vegetation reconstructions [49] and glacial extents [50]. Therefore, ROAD represents a valuable source of data for classical spatial analysis and modeling, spatial network analysis, agent based modeling, ecological niche modeling, spatial diffusion modeling and the growing number of methods fueled by geospatial machine learning and artificial intelligence.

### 5.3 Sustainable development and long-term perspectives

As a long-term project, ROCEEH had to respond to general trends in scientific data infrastructures and science itself [51]. In the end, we developed ROAD along a trajectory towards open, FAIR [3] and linked data. When the research center launched in 2008, data sustainability played a minor role, and the database was conceived as an island solution to address the specific research questions of the ROCEEH research center. In subsequent years, milestones such as the Panton Principles [52], the FAIR Principles [7], the Horizon 2020 guidelines for research funding, and the UNESCO recommendations on open science [53] changed the research landscape significantly. Open science—and with it the possibility to synthesize existing data [25]—went from being a mere suggestion to a categorical requirement.

In ROAD, these developments are reflected in the creation of API interfaces and through networking with other databases, such as NEOTOMA [16] and NQMDB. Starting in 2018, we targeted specific measures to improve compliance with the FAIR principles. We replaced our older, ambiguous data use policy with a standard Creative Commons license (CC BY-SA 4.0), making the terms of use easy to understand and transparent for both data providers and users (S3 File). To improve interoperability, we established another semantic format (RDF) parallel to the original relational database. As a result, it became possible to maintain data as RDF triples. This allowed the formulation of queries using SPARQL query language, which is used widely in the digital humanities [54, 55].

These steps vaulted ROAD into a position where its data became interoperable and ready for Linked Open Data [56, 57]. Thanks to the RDF format, it became possible to map ROAD data into the ontology of other large databases. For example, ARIADNEplus aggregates data, offering an online resource containing more than three million datasets from diverse archaeological sources.

The use of RDF also required us to standardize many aspects of our own dataset. First, we learned to map the information in ROAD using ARIADNE's semantics, called the AO-CAT schema. Second, our existing cultural periods and technocomplexes were defined and linked with properties such as spatial coverage and temporal extent in PeriodO [58]. Third, cultural finds were represented by entries in the Getty Art & Architecture Thesaurus. By following these steps, we made ROAD data available in ARIADNE's standardized scheme. Thus, ROAD became more accessible through the indexing afforded by the ARIADNE portal, which functions as a spatio-temporal search engine. These efforts served to increase ROAD's findability considerably.

Our next goal is to store the contents of the database in a long-term repository with persistent identifiers such as DOIs. Such solutions are meant to preserve the data structure, as well as the information collected in ROAD, in a sustainable way. Nonetheless, the functionality of ROADWeb—developed over the past 15 years—and especially the ability to keep its content up-to-date will diminish when the project ends in 2027. The database includes supporting software such as user interfaces and analytical tools that would no longer be maintained. One way to keep the content of ROAD accessible would be to create an open-access catalog of site summary data sheets (S4 File) from every locality. While these PDFs provide an overview of each site, they do not provide access to the datasets themselves.

New discoveries in prehistory are increasing constantly [59]. The reassessment of existing finds, the reevaluation of stratigraphies and chronologies [60–62], and the reframing of biological and cultural taxonomies [63] continue to change the research landscape. Thus, by storing ROAD in a long-term repository, we would merely create a modern version of an atlas with its final publishing date. While such a book is useful for quick reference, it eventually collects dust on a shelf. Such a solution would deprive ROAD of its digital potential, as this database represents:

a. an open source of global knowledge about humanity's deep heritage including difficult-to-access information due to language, distribution, or paywall limits of the original publications;

b. an analytical tool that can be mastered without expertise in data processing or programming;

c. a framework to explore data about the first three million years of human evolution in an interdisciplinary, diachronic, and transcontinental way, opening up insight into hidden patterns, interrelations, differences and pathways for new research questions.

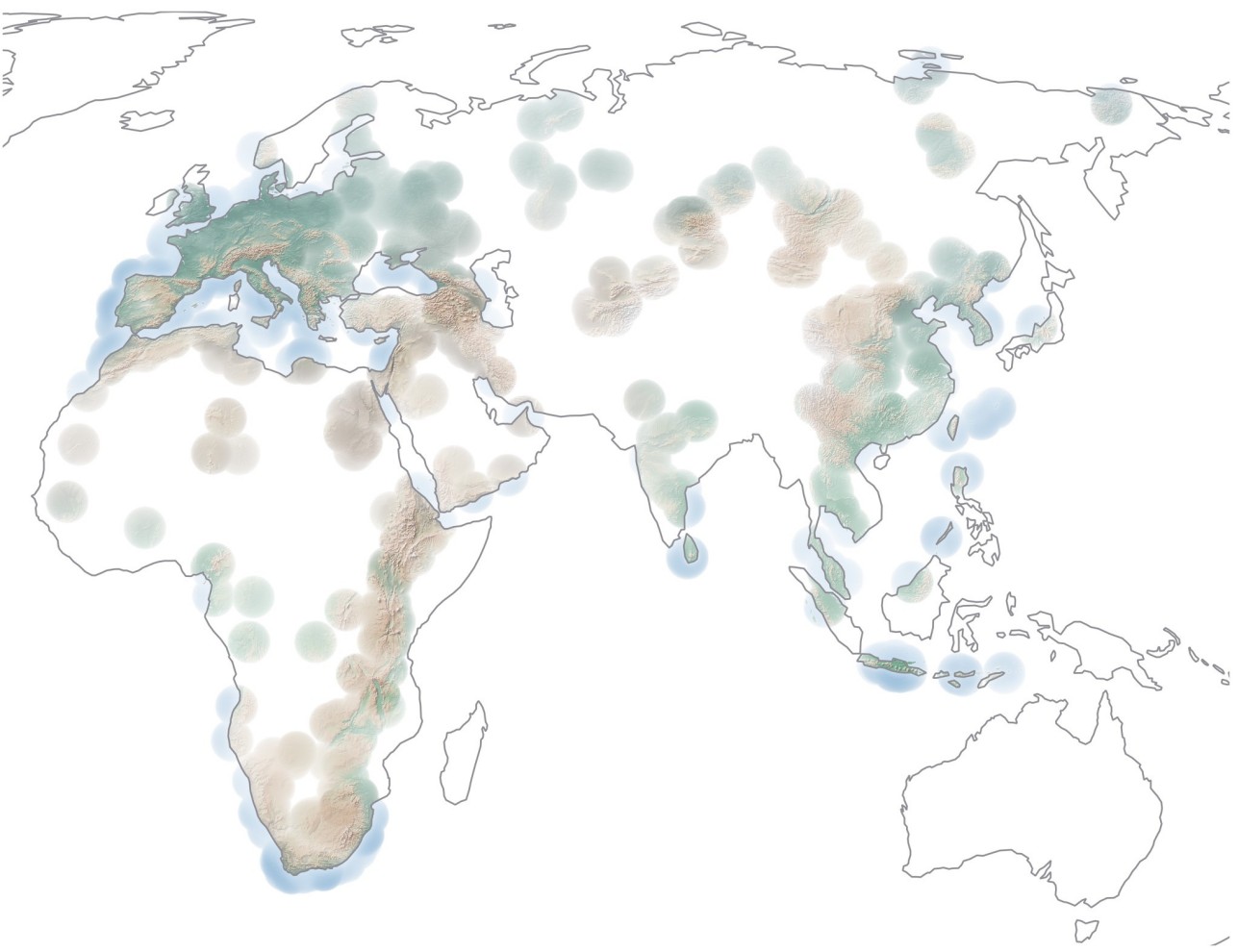

**Fig 22. Missing data—Assembling the pieces of the puzzle.** Despite an intensive focus on data entry, the majority of the prehistoric world remains uncharted, as shown by the blank areas of this map. Visualization based on a Kernel Density Estimate of ROAD assemblages. Sites with higher assemblage densities appear more intense in color. Note that Australia is not within the database's scope. Map credit: Made with Natural Earth (public domain).

Therefore, it is our aim to use the remaining years of ROCEEH to find a host institution and basic funding to insure, at a minimum, its existing functionality and the maintenance of data. In the best case scenario, the updating of data and development of functional tools would also continue.

## 6. Conclusion

The ROCEEH Out of Africa Database is a large-scale data management tool that enables the collection, management, analysis and presentation of data for researchers studying the deep history of early humans and their environments. It is the largest database of its kind, covering Africa and Eurasia with a time depth from 3,000,000 to 20,000 years ago. Its database model mirrors the ROCEEH project's interdisciplinary approach and facilitates a holistic view of sites by combining archaeological, paleoanthropological, paleontological, paleobotanical and paleo-geographical information. Its contents are compiled from a variety of sources, including an extensive body of literature that makes publications in various languages and from different

regions accessible. ROCEEH designed several analytical tools to query, aggregate and visualize data and help users distill knowledge from this vast amount of information. Compliance with the FAIR principles makes the database suitable for a wide range of user-friendly applications.

We see ROAD as an essential tool for scientific exploration, whether researchers want an overview of individual sites or require big data for intensive computational analysis. Furthermore, the database serves as a blueprint for integrated data collection in interdisciplinary research as well as digitalization and standardization of published legacy data. It is a tool for the public audience to explore the prehistory of humanity, with a track record of museum exhibitions, literature in popular science [64] and even computer games [65].

Nonetheless, our results show that our knowledge of human evolution, as covered by ROAD, is highly fragmented (Fig 22). We do not claim that ROAD is "complete", as new data are constantly being created and some regions have yet to be fully entered. Still, it becomes apparent that scientific research of our past is strongly biased, concentrated on just a few regions. Completing this picture will be a collaborative task for the entire scientific community. Collected data must be made FAIR through machine-readability and common data standards, shared vocabularies and infrastructure. Furthermore, sources that are often ignored by Western archives need to be recognized. Finally, funding agencies should be open to risky investments to approach understudied regions and support and develop local scientific expertise.

## Supporting information

**S1 Fig. The ROAD database model.** We developed this data model using the visual tool MySQL Workbench to create this Entity Relationship Diagram. The model depicts all tables and attributes in ROAD and shows primary and foreign keys with their relationships within the database.
(TIF)

**S1 Table. Names and hierarchy of all scientific tables in ROAD.** This list of the tables used for data entry is divided according to the four scientific disciplines. Tables are listed in the order they would be completed while entering data. Note that archaeological finds always have an archaeological layer that contains the cultural designation, while the other categories do not. Colors are coded to the disciplines, although entries in black are shared by all disciplines.
(DOCX)

**S1 File. The ROAD table descriptions.** This guide provides detailed definitions of all tables and their attributes, specifications for data entry and examples of possible entries.
(PDF)

**S2 File. The ROAD manual.** The manual provides general information about the database, describes how to write queries, illustrates how to use the Map Module to visualize data.
(PDF)

**S3 File. The user agreement.** This document specifies the responsibilities of a user and serves as an application to set up a user account.
(PDF)

**S4 File. A site summary data sheet.** This feature makes important information about a locality available through the click of a button. The resulting PDF, for example, for Aghitu-3 Cave in Armenia demonstrates the utility of this function.
(PDF)

## Acknowledgments

This manuscript benefited from the contributions of ROCEEH's many colleagues and associates through stimulating discussions at international workshops, conferences and congresses as well as through personal meetings and correspondence. Numerous colleagues in many countries contributed lists of sites, articles and their own personal datasets to be included in ROAD. We thank the many users of ROAD who have pointed out mistakes and inconsistencies in the data, which we have faithfully tried to correct and improve. Most important of all, we are grateful to the dozens of student research assistants at the Universities of Frankfurt and Tübingen who have entered data into ROAD since 2009. Without their hard work and perseverance, ROAD could never have taken its current shape. Finally, we appreciate the thoughtful comments of Emily Hallinan and an anonymous reviewer, whose additions helped improve the quality of this paper.

## Author Contributions

**Conceptualization:** Andrew W. Kandel, Christian Sommer, Zara Kanaeva, Michael Bolus, Angela A. Bruch, Miriam N. Haidle, Christine Hertler, Maria Malina, Volker Hochschild.

**Data curation:** Andrew W. Kandel, Christian Sommer, Zara Kanaeva.

**Formal analysis:** Andrew W. Kandel, Christian Sommer, Zara Kanaeva, Michael Bolus, Angela A. Bruch, Miriam N. Haidle, Christine Hertler.

**Funding acquisition:** Volker Hochschild, Volker Mosbrugger, Friedemann Schrenk, Nicholas J. Conard.

**Investigation:** Andrew W. Kandel, Christian Sommer, Zara Kanaeva, Michael Bolus, Angela A. Bruch, Miriam N. Haidle, Christine Hertler, Maria Malina, Volker Hochschild.

**Methodology:** Andrew W. Kandel, Christian Sommer, Zara Kanaeva, Michael Bolus, Angela A. Bruch, Miriam N. Haidle, Christine Hertler, Maria Malina, Michael Märker, Volker Hochschild.

**Project administration:** Claudia Groth, Miriam N. Haidle, Julia Heß, Maria Malina.

**Software:** Christian Sommer, Zara Kanaeva.

**Validation:** Christian Sommer, Zara Kanaeva.

**Visualization:** Andrew W. Kandel, Christian Sommer, Maria Malina.

**Writing – original draft:** Andrew W. Kandel, Christian Sommer, Zara Kanaeva, Michael Bolus, Angela A. Bruch, Miriam N. Haidle, Christine Hertler, Volker Hochschild.

**Writing – review & editing:** Andrew W. Kandel, Christian Sommer, Zara Kanaeva, Michael Bolus, Angela A. Bruch, Miriam N. Haidle, Christine Hertler, Maria Malina, Michael Märker, Volker Hochschild, Friedemann Schrenk, Nicholas J. Conard.

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
