## [Decision Letter · Decision Letter 0]

3 May 2023

PONE-D-23-06596The ROCEEH Out of Africa Database (ROAD): A large-scale research database serves as an indispensable tool for human evolutionary studiesPLOS ONE

Dear Dr. Kandel,

Thank you for submitting your manuscript to PLOS ONE. After careful consideration, we feel that it has merit but does not fully meet PLOS ONE’s publication criteria as it currently stands. Therefore, we invite you to submit a revised version of the manuscript that addresses the points raised during the review process.

We look forward to receiving your revised manuscript.

Kind regards,

Enza Elena Spinapolice, Ph.D

Academic Editor

PLOS ONE

3. We note that Figures 12,14,15,19,21 and 22 in your submission contain [map/satellite] images which may be copyrighted. All PLOS content is published under the Creative Commons Attribution License (CC BY 4.0), which means that the manuscript, images, and Supporting Information files will be freely available online, and any third party is permitted to access, download, copy, distribute, and use these materials in any way, even commercially, with proper attribution. For these reasons, we cannot publish previously copyrighted maps or satellite images created using proprietary data, such as Google software (Google Maps, Street View, and Earth). For more information, see our copyright guidelines: http://journals.plos.org/plosone/s/licenses-and-copyright.

a. You may seek permission from the original copyright holder of Figures 12,14,15,19,21 and 22 to publish the content specifically under the CC BY 4.0 license. 

Reviewers' comments:

Reviewer's Responses to Questions

**Comments to the Author**

1. Is the manuscript technically sound, and do the data support the conclusions?

Reviewer #1: Yes

Reviewer #2: Yes

2. Has the statistical analysis been performed appropriately and rigorously? 

Reviewer #1: Yes

Reviewer #2: Yes

3. Have the authors made all data underlying the findings in their manuscript fully available?

Reviewer #1: Yes

Reviewer #2: Yes

4. Is the manuscript presented in an intelligible fashion and written in standard English?

Reviewer #1: Yes

Reviewer #2: Yes

5. Review Comments to the Author

Reviewer #1: The manuscript presents an overview of and guide to using the ROAD database, developed by the ROCHEEH research center over the past 15 years. The concept behind the project was to compile data on archaeological occurrences between 3 million and 20,000 years ago to enable scientists to conduct large-scale analyses of spatial, temporal and cultural aspects of human expansions across Africa and Eurasia. The database has a free-to-use online search facility as well as allowing extended search capabilities to registered users. The manuscript details the rationale behind the database, its structure and features, and its application to research-questions. Finally, the future of the database is considered and the advantages and drawbacks of alternative storage methods and maintenance of the data are presented. As emphasized throughout the manuscript, strategies for effective data management and open access are a primary concern in current research design. Both the structure, replicability and accessibility of the database, and the project’s vision for the longer-term should serve as a model for similar initiatives across the sciences.

Since this is not a standard research article, I have few comments other than to commend the authors on creating a valuable open-source tool for the scientific community and presenting such user-friendly guides and features. Two very minor queries are:

Sect. 2.3, lines 273-275: I understand the need to score sampling/representativeness numerically in the database structure but is there anywhere for details to be given for individual cases or explain how scores are assigned?

Lines 284-5: “In general, archaeological finds are entered as groups of finds and not as single artifacts.”

Are symbolic artifacts given the same treatment? – the implication by nature of the artifact type is that these are entered individually. Add a few words to clarify.

Reviewer #2: The paper addresses the crucial issue of the utilisation of big data in archaeology, presenting the ROAD database of the ROCEEH team. The ROAD database is the container of a large masses of data covering many aspects of the Role of Culture in Early Expansions of Humans. The ROCEEH research group transformed 5020 titles into machine-readable data, ensuring their interoperability with others large dataset.

The authors clearly describe the structure of the database. It appears to be very well designed and is able to manage and relate heterogeneous data. Great attention has been paid to managing the different interpretations of the same data, guaranteeing comprehensive information to users. The software solutions adopted (e.g. PostgreSQL as DBMS) are appropriate, and the web user interface is intuitive and easy to use. Of note is the programming of APIs that allow ROAD to dialogue with other applications.

Minor issues

1 It is not clear how human remains are stored in the case of human bodies (and not a single item).

2 Stone artefacts tables raw_materials, typology, technology, function have a *list field. What kind of data do these fields store?

3 Statement between line 407 and 410 is not very clear.

4 It is not clear how the validation workflow works. Table 1 shows that quality control is guaranteed by user of level 4 (ROCEEH team members), but modalities are not specified in the text.

5 It is not clear stage of the process reached in incorporating ROAD into the Semantic Web.

6. PLOS authors have the option to publish the peer review history of their article (what does this mean?). If published, this will include your full peer review and any attached files.

Reviewer #1: **Yes: **Emily Hallinan

Reviewer #2: No

---

## [Author Response · Author response to Decision Letter 0]

20 May 2023

Copied from "Response to Reviewers"

Comments to the Author

We thank the reviewers and journal editors for their thoughtful comments and complied with their wishes as described below. In addition to track changes, we highlighted those changes in yellow to make it easier for the reviewers to find those changes. 

Please note that “new line” numbers always refer to the “Revised Manuscript with Track Changes”. 

We made some minor changes: 

• moved a sentence and deleted another for clarity (new lines 406-408, 412-413). 

• added a sentence to the section entitled “Acknowledgements” (new lines 1042-1043) to reinforce our thanks to the reviewers. 

Done 

The repository can now be accessed at: https://doi.org/10.5281/zenodo.7669784. With the opening of the repository link, the temporary link to our cloud server is no longer accessible. We noted these changes in the section entitled “Data management” (new lines 1024-1026), and in the new cover letter. 

3. We note that Figures 12,14,15,19,21 and 22 in your submission contain [map/satellite] images which may be copyrighted. All PLOS content is published under the Creative Commons Attribution License (CC BY 4.0), which means that the manuscript, images, and Supporting Information files will be freely available online, and any third party is permitted to access, download, copy, distribute, and use these materials in any way, even commercially, with proper attribution. For these reasons, we cannot publish previously copyrighted maps or satellite images created using proprietary data, such as Google software (Google Maps, Street View, and Earth). For more information, see our copyright guidelines: http://journals.plos.org/plosone/s/licenses-and-copyright.

We revised the figures according to the journal’s specifications. For Figures 12 and 14, we changed the maps from Google Maps to OpenStreetMap and hope this is satisfactory. We also added “Map credit: OpenStreetMap” to those captions. For Figures 15, 19, 21 and 22 we already used open access maps and added “Map credit: Made with Natural Earth (public domain)” to those captions, as suggested by the creators of these maps. 

Done 

5. Review Comments to the Author

Reviewer #1:

The manuscript presents an overview of and guide to using the ROAD database, developed by the ROCHEEH research center over the past 15 years. The concept behind the project was to compile data on archaeological occurrences between 3 million and 20,000 years ago to enable scientists to conduct large-scale analyses of spatial, temporal and cultural aspects of human expansions across Africa and Eurasia. The database has a free-to-use online search facility as well as allowing extended search capabilities to registered users. The manuscript details the rationale behind the database, its structure and features, and its application to research-questions. Finally, the future of the database is considered and the advantages and drawbacks of alternative storage methods and maintenance of the data are presented. As emphasized throughout the manuscript, strategies for effective data management and open access are a primary concern in current research design. Both the structure, replicability and accessibility of the database, and the project’s vision for the longer-term should serve as a model for similar initiatives across the sciences.

Since this is not a standard research article, I have few comments other than to commend the authors on creating a valuable open-source tool for the scientific community and presenting such user-friendly guides and features. Two very minor queries are:

Sect. 2.3, lines 273-275: I understand the need to score sampling/representativeness numerically in the database structure but is there anywhere for details to be given for individual cases or explain how scores are assigned?

We added text to the first paragraph of Section 2.3 to explain this (new lines 267-273). Please also note that full explanations of each table and every attribute are presented in the Supplementary Materials (S1 File. The ROAD Table Descriptions). We added a sentence about this to Section 2.1 (new lines 180-181).

Lines 284-5: “In general, archaeological finds are entered as groups of finds and not as single artifacts.” Are symbolic artifacts given the same treatment? – the implication by nature of the artifact type is that these are entered individually. Add a few words to clarify.

Like the lithics, symbolic artifacts are also grouped together. This grouping is based on their interpretation (i.e. art, music, ornament) and their material (e.g., stone, bone, shell, etc.) This has been clarified in the text in Section 2.3.1 (new lines 338-341).

Reviewer #2:

The paper addresses the crucial issue of the utilisation of big data in archaeology, presenting the ROAD database of the ROCEEH team. The ROAD database is the container of a large masses of data covering many aspects of the Role of Culture in Early Expansions of Humans. The ROCEEH research group transformed 5020 titles into machine-readable data, ensuring their interoperability with others large dataset.

The authors clearly describe the structure of the database. It appears to be very well designed and is able to manage and relate heterogeneous data. Great attention has been paid to managing the different interpretations of the same data, guaranteeing comprehensive information to users. The software solutions adopted (e.g. PostgreSQL as DBMS) are appropriate, and the web user interface is intuitive and easy to use. Of note is the programming of APIs that allow ROAD to dialogue with other applications.

Minor issues

1 It is not clear how human remains are stored in the case of human bodies (and not a single item).

Human remains are usually entered as individual finds. The exception is when they belong to a single human body, for example as part of a burial, or as fragments of remains that can be attributed to a single individual. We have clarified this in Section 2.3.2 (new lines 379-384).

2 Stone artefacts tables raw_materials, typology, technology, function have a *list field. What kind of data do these fields store?

We specify the lists in the four relevant paragraphs of Section 2.3.1 (new lines 300-302, 315-316, 323-326, 331-334) and hope this helps. 

3 Statement between line 407 and 410 is not very clear.

We hope our changes to the caption of Fig. 10 help to make this clearer (new lines 426-447). We eliminated the sentence about the foreign keys, as this is a technical detail that is not essential for understanding the connections.

4 It is not clear how the validation workflow works. Table 1 shows that quality control is guaranteed by user of level 4 (ROCEEH team members), but modalities are not specified in the text.

In Section 3.2 first we added a paragraph about the abilities of the four groups, as presented in Table 1 (new lines 507-513). This segues into a paragraph about how we implement quality control in ROAD (new lines 514-519). 

5 It is not clear stage of the process reached in incorporating ROAD into the Semantic Web.

In Section 3.5.4 (new lines 641-650) we go into more detail about the Semantic Web, the SPARQL endpoint and the status of our RDF exports.

---

## [Decision Letter · Decision Letter 1]

20 Jul 2023

The ROCEEH Out of Africa Database (ROAD): A large-scale research database serves as an indispensable tool for human evolutionary studies

PONE-D-23-06596R1

Dear Dr. Kandel,

We’re pleased to inform you that your manuscript has been judged scientifically suitable for publication and will be formally accepted for publication once it meets all outstanding technical requirements.

Kind regards,

Enza Elena Spinapolice, Ph.D

Academic Editor

PLOS ONE

Additional Editor Comments (optional):

Reviewers' comments:

Reviewer's Responses to Questions

**Comments to the Author**

1. If the authors have adequately addressed your comments raised in a previous round of review and you feel that this manuscript is now acceptable for publication, you may indicate that here to bypass the “Comments to the Author” section, enter your conflict of interest statement in the “Confidential to Editor” section, and submit your "Accept" recommendation.

Reviewer #2: All comments have been addressed

2. Is the manuscript technically sound, and do the data support the conclusions?

Reviewer #2: Yes

3. Has the statistical analysis been performed appropriately and rigorously? 

Reviewer #2: N/A

4. Have the authors made all data underlying the findings in their manuscript fully available?

Reviewer #2: Yes

5. Is the manuscript presented in an intelligible fashion and written in standard English?

Reviewer #2: Yes

6. Review Comments to the Author

Reviewer #2: The authors answered all comments in full. The manuscript needs no further comments and can be published

7. PLOS authors have the option to publish the peer review history of their article (what does this mean?). If published, this will include your full peer review and any attached files.

Reviewer #2: No

---

## [Editor Report · Acceptance letter]

24 Jul 2023

PONE-D-23-06596R1 

The ROCEEH Out of Africa Database (ROAD): A large-scale research database serves as an indispensable tool for human evolutionary studies 

Dear Dr. Kandel:

I'm pleased to inform you that your manuscript has been deemed suitable for publication in PLOS ONE. Congratulations! Your manuscript is now with our production department. 

Kind regards, 

on behalf of

Dr. Enza Elena Spinapolice 

Academic Editor

PLOS ONE